# Forestalled phase separation as the precursor to stripe order

Aritra Sinha ● ✉ & Alexander Wietek ●

Stripe order is a prominent feature in the phase diagram of the cuprate superconductors and has been established as the lowest-energy state of the two-dimensional Fermi-Hubbard model in relevant regimes. As temperature rises, stripes and superconductivity give way to the interesting strange-metal and pseudogap regimes. Here we investigate a crucial aspect of these regimes using numerical simulations of the square lattice Hubbard model. Using infinite projected entangled pair states with purification in the thermodynamic limit, we show that the thermodynamic charge susceptibility develops a broad maximum above the stripe regime at filling near $n = 0.90$, that strengthens upon cooling. Minimally entangled typical thermal states on finite cylinders attribute this enhancement to the emergence of large, fluctuating charge clusters, while the charge susceptibility remains finite. Upon further cooling, the cluster sizes stop broadening and lock to the stripe wavelength, indicating that interaction-driven clustering organizes the intermediate-temperature landscape but is ultimately forestalled by stripe order at low temperature.

Phase separation (PS) in high-$T_c$ superconductors (HTSCs) refers to an underlying tendency toward coexistence of hole-rich metallic regions and hole-poor antiferromagnetic domains[1,2]. Experimentally, such inhomogeneities were observed early on in $La_2CuO_{4+\delta}$[3] and later in stripe-ordered $La_{1.6-x}Nd_{0.4}Sr_xCuO_4$[4]. Beyond cuprates[5-7], PS tendencies also appear in iron-based chalcogenides[8,9], doped manganites[10], and nickelates[11]; see ref. 12 for a recent review.

Theoretical scenarios link PS, charge-density modulations, and superconductivity[1,2]. Early mean-field theories proposed that doped carriers cluster into metallic networks pivotal for superconductivity[1,13,14]. Concepts of charge density waves (CDW) and stripe order[15-18] emerged in tandem with ideas about electronic inhomogeneities[19]. Emery and Kivelson's frustrated PS scenario[20] posited that long-range Coulomb interactions could prevent full macroscopic segregation, favoring CDW-like patterns. Recent work on Hubbard-Holstein and related models[21-23] further underscores the importance of electron-lattice interactions in facilitating phase separation.

The square-lattice two-dimensional (2D) Fermi-Hubbard model (FHM)[24] is central to understanding high-Tc superconductivity[25-27]. Its phase diagram, restricted to nearest-neighbor (NN) hopping, has been debated extensively[28,29], but a growing consensus points to stripe order at slight doping under cuprate-relevant conditions[30-38]. The

pseudogap and strange-metal regimes, prominent in cuprates above the stripe order and superconducting phases, have driven intense scrutiny of the Hubbard model at finite temperature[39-44]. Notably, a cellular dynamical mean-field theory (CDMFT) study[40] tied pseudogap formation to short-range spin correlations, and diagrammatic Monte Carlo simulations[44] suggest a predominantly spin-driven origin. Another study[43] using CDMFT reinforced the spin-driven origin of the pseudogap, while highlighting that charge inhomogeneities may play some role.

In this work, we combine two advanced tensor-network approaches–minimally entangled typical thermal states (METTS)[45,46] built on matrix product states (MPS)[47,48], and purification[49] with infinite projected entangled pair states (iPEPS)[50,51]–to study both quasi-1D cylindrical geometry and the infinite 2D limit, charting the finite-temperature evolution of the square-lattice Fermi-Hubbard model at strong coupling. We find (i) a broad maximum of the thermodynamic charge susceptibility $\chi_{charge} = \partial n/\partial \mu$ near $n \approx 0.90$ that intensifies upon cooling, (ii) an intermediate-temperature regime of large, fluctuating hole clusters coexisting with enhanced $(\pi, \pi)$ spin correlations, and (iii) a crossover to static stripe order at lower temperature where cluster sizes lock to the stripe wavelength. Taken together, these results identify a forestalled phase-separation regime–interaction-driven

Max Planck Institute for the Physics of Complex Systems, Nöthnitzer Strasse 38, Dresden, Germany. ✉e-mail: asinha@pks.mpg.de

charge aggregation without a divergent compressibility–that is ultimately preempted by stripe order.

## Results

### Charge susceptibility in the thermodynamic limit

In this section, we implement the purification method[52] via fermionic iPEPS[51] following ref. [53]. Here, we use the particle-hole symmetric form of the FHM,

$$
H = -\sum_{\langle i,j \rangle, \sigma} t \left( c_{i\sigma}^\dagger c_{j\sigma} + c_{j\sigma}^\dagger c_{i\sigma} \right) \\
+ \sum_i U \left( n_{i\uparrow} - \frac{1}{2} \right) \left( n_{i\downarrow} - \frac{1}{2} \right) - \sum_i \mu\, n_i, \tag{1}
$$

where $t$ represents the NN hopping amplitude and $U > 0$ denotes the on-site Coulomb repulsion. Here, $c_{i\sigma}^\dagger$ ($c_{i\sigma}$) is the creation (annihilation) operator for an electron with spin $\sigma$ at site $i$, and $n_{i\sigma} = c_{i\sigma}^\dagger c_{i\sigma}$ is the number operator for electrons of spin $\sigma$ at site $i$. The summation $\langle i, j \rangle$ runs over NN sites on the square lattice. With this convention, half-filling sits at $\mu = 0$. In this article, we set $t = 1$. We control the filling $n$ by changing the chemical potential $\mu$.

In this grand-canonical formalism, the system exchanges particles with a reservoir, and the particle density $n$ adjusts to reach equilibrium, leading to a unique equilibrium density for a given chemical potential. The charge susceptibility $\chi_{\text{charge}} = \frac{\partial n}{\partial \mu}$ quantifies how sensitive the particle density is to changes in the chemical potential. The sketch Fig. 1b illustrates three scenarios with $\chi_{\text{charge}}$ versus $\mu$ plots and density $n$ versus $\mu$ plots (inset). Scenario **I** (pink curve) shows PS at ground state for some $\mu = \mu_c$ with a sharp discontinuity in the $n$ vs $\mu$ plot and a divergent peak in $\chi_{\text{charge}}$, suggesting an enhanced susceptibility. This signature has been used to provide evidence for PS in iPEPS studies of fermions on square and honeycomb lattices[51,54]. Because the iPEPS unit cell selected here is small and translationally invariant, actual

inhomogeneous states cannot be represented; one can detect PS tendencies only via peaks in $\chi_{\text{charge}}$. Scenario **II** (green curve) describes a high-temperature state with thermal fluctuations that smooth the $n$ vs $\mu$ curve and keep $\chi_{\text{charge}}$ relatively flat. There is also a scenario **III** (orange curve), which demonstrates a finite-temperature crossover. Here, the $n$ vs $\mu$ curve exhibits a noticeable change in curvature without a discontinuous jump. The charge susceptibility $\chi_{\text{charge}}$ in this case shows a modest peak, signifying an increased, but finite, susceptibility to variations in $\mu$. As we show below, in the Hubbard model, this can result from statistically fluctuating hole clusters that mimic phase separation on a smaller scale without achieving full separation. Cartoon illustrations of Scenario **I** and **III** for the Fermi-Hubbard model are presented in Fig. 1a, with Scenario **I** depicted on the left and Scenario **III** on the right. In the bottom panel of Fig. 1, we show iPEPS results for an infinite square lattice at $U = 10$ – (c) plots the filling $n$ versus chemical potential $\mu$, while (d) displays the corresponding charge susceptibility. At high temperature ($T = 4$), the system follows the smooth behavior of Scenario II (no PS). As $T$ is lowered, the $n(\mu)$ curves develop an increasingly sharp inflection, and $\chi_{\text{charge}}(\mu)$ peaks become more pronounced, signaling a crossover towards Scenario III. In Fig. 1e, we replot the same charge susceptibility data against the density $n$ instead of $\mu$ and find that the maximum occurs in a broad region surrounding density $n \approx 0.91$. Importantly, plaquette CDMFT studies of the square-lattice Hubbard model at strong coupling[55] show that the van Hove singularity (defined via the maximum in the local density of states at the Fermi level) appears at much larger doping than our observed susceptibility peak near doping $1 - n \approx 0.09$ for comparable Coulomb strength $U$. This strongly suggests that our result does not arise from a non-interacting band-structure effect, but instead reflects genuine strong-coupling physics associated with Mottness. This behavior is consistent with findings from dynamical cluster approximation (DCA)[56,57] for the Hubbard model with nearest-neighbor hopping only ($t' = 0$), which show that $\chi_{\text{charge}}$ stays finite at nonzero temperature yet

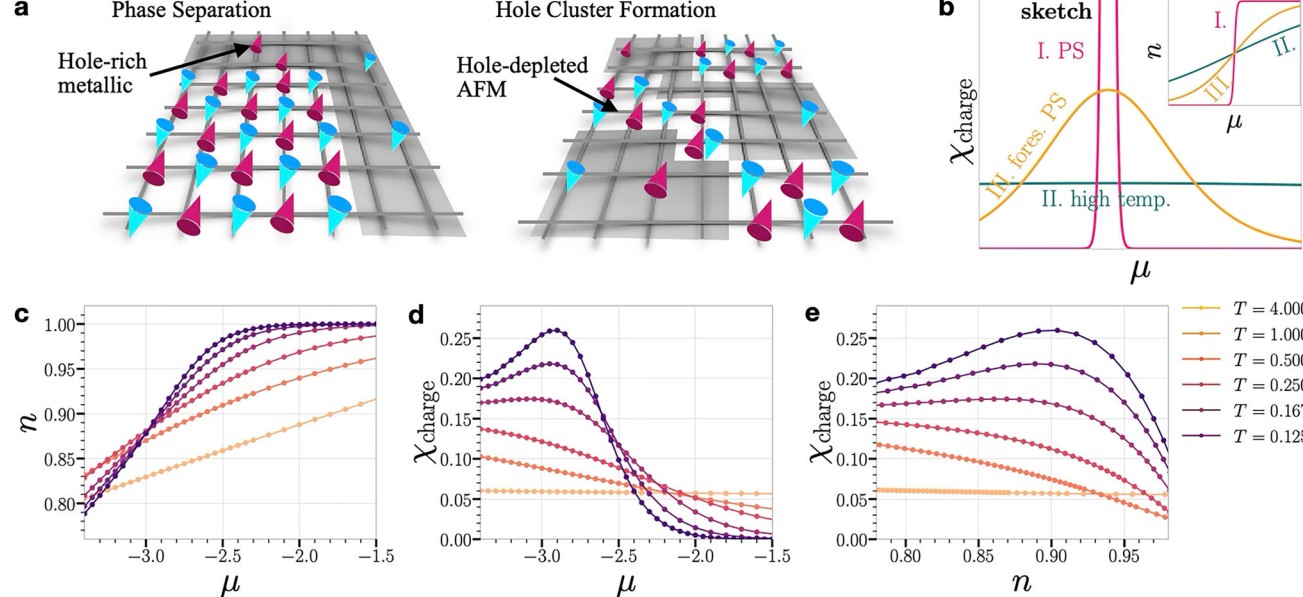

**Fig. 1 | Charge susceptibility. a** Cartoon illustrations of phase separation in the Fermi-Hubbard model on a square lattice. The left panel shows complete phase separation, with a hole-rich phase (shaded gray) and an antiferromagnetic domain (unshaded), with red and blue cones representing spin-up and spin-down polarization of the electrons. The right panel depicts hole clustering, where thermal fluctuations prevent complete phase separation, resulting in clusters of holes accompanied by AFM domains. **b** Schematic of charge susceptibility $\chi_{\text{charge}} = \frac{\partial n}{\partial \mu}$ vs. chemical potential $\mu$, depicting three possible scenarios: **I** (pink curve) indicates sharp phase separation at the ground state, with a discontinuity in $n$ (inset) and a

divergent peak in $\chi_{\text{charge}}$; **II** (green curve) represents high-temperature behavior with smoothed $n$ vs. $\mu$ and flat $\chi_{\text{charge}}$; and **III** (orange curve) displays near-critical behavior at finite temperature, with a small peak in $\chi_{\text{charge}}$. We refer to this scenario as forestalled phase separation (labeled `fores. PS' in the sketch). The bottom panel shows iPEPS simulations done on an infinite square lattice at $U = 10$. In (**c**), we plot $n(\mu)$ vs $\mu$, and in (**d**), we show $\chi_{\text{charge}}$ as a function of $\mu$. We observe a shift from Scenario **II** to **III** for the latter as the temperature $T$ decreases from $T = 4.000$ to $T = 0.125$. Additionally, in (**e**), we plot $\chi_{\text{charge}}$ vs. $n$, which highlights the values of the density where $\chi_{\text{charge}}$ is maximal.

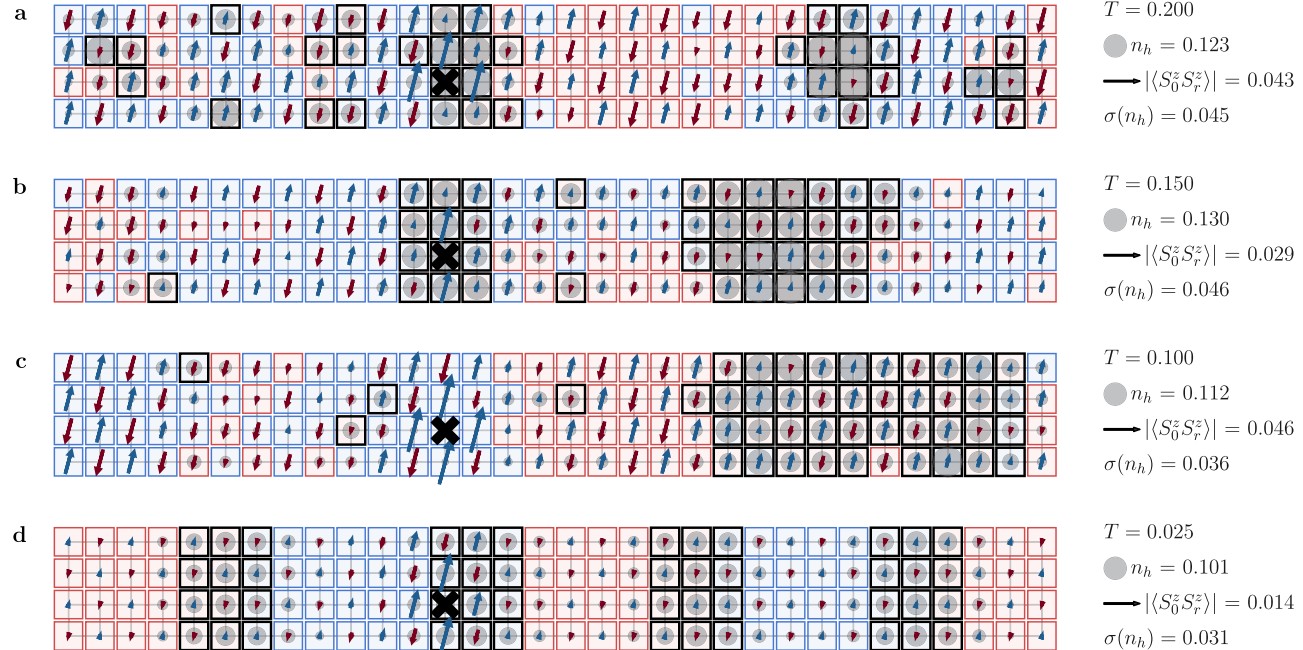

**Fig. 2 | METTS snapshots.** Visualizations of hole density and spin correlations for a typical METTS state $|\psi_i\rangle$ in the Hubbard model with $U=10$ at electron density $n = 0.9375$ on a $32 \times 4$ cylinder. Gray circles, with diameters proportional to hole densities $n_h(r) = 1 - \langle n_r \rangle = 1 - \langle \psi_i | n_r | \psi_i \rangle$, highlight variations in local charge distribution. Spin correlations are shown by arrows (red for negative and blue for positive) of lengths proportional to the correlation strengths $\langle \psi_i | S_0^z \cdot S_r^z | \psi_i \rangle$; the black cross denotes the origin $r = 0$ of the spin correlator. Red and blue squares mark the sign and spatial structure of staggered spin correlations, $(-1)^{x+y} \langle \psi_i | S_0^z \cdot S_r^z | \psi_i \rangle$, thereby assigning the same color to each site within a given antiferromagnetic domain. Black-bordered sites indicate hole densities exceeding a specified threshold (see Eq. (4)); neighboring black-bordered sites define the clusters $\mathcal{C}$ in Eq. (7). Panels (**a–d**) from top to bottom show snapshots at temperatures $T = 0.200, 0.150, 0.100,$ and $0.025$, respectively. The right-hand scale uses fixed-size icons (constant reference circle radius and arrow length across all panels) for visual comparison and $\sigma_{n_h}$, the standard deviation of $n_h$ within a snapshot. At the lowest temperature ($T = 0.025$), panel (**d**) shows a stripe pattern, i.e., a charge-density wave intertwined with a spin-density wave. The antiferromagnetic correlations undergo a $\pi$-phase shift across the domain wall that runs through the middle of clusters of dominant-size $m = |\mathcal{C}| = 12$ sites.

increases as $T \to 0$. With a finite $t'$, however, the model was shown to undergo a first-order phase separation transition at finite $T$, terminating at a second-order critical point where $\chi_{\text{charge}}$ diverges. This critical point aligns with a quantum critical point ($T = 0$) $t' = 0$, separating pseudogap and Fermi-liquid regions. Sordi et al.[58] connect this behavior to the Widom line, a line of maxima in $\chi_{\text{charge}}$ that extends from the QCP, and which represents a thermodynamic crossover boundary, organizing the phase space and marking the onset of the pseudogap phase at a characteristic temperature $T^*$. Concurrent work on a triangular-lattice kinetic antiferromagnet reports true finite-temperature phase separation and a charge-magnon liquid[59], complementary to our square-lattice Hubbard results, where clustering is ultimately preempted by stripe order.

## Hole clustering in METTS snapshots

Simulations with iPEPS at finite temperature in the grand-canonical ensemble typically result in a uniform particle density across the system due to the choice of a translationally invariant ansatz. Studies done in the canonical ensemble using METTS[45,46] fix the total number of particles. As a result, charge inhomogeneities can readily be observed. Here, we follow the methods outlined in ref. 60. We conducted our simulations primarily on cylinders of width $W = 4, 6$ (periodic) and length $L \le 40$ (open). Using the same on-site Coulomb repulsion ($U = 10$), we set the electron filling to $n = 0.9375$. We chose this filling as it offered favorable convergence properties, while detailed studies in Supplementary Section IV confirm that our conclusions are valid across a broad range of fillings. The METTS algorithm, a Markov-chain Monte Carlo method, simulates thermal states via iterative imaginary-time evolution of product states[45,46]. The

evolution process involves creating METTS snapshots $|\psi_i\rangle$,

$$|\psi_i\rangle = \frac{1}{\sqrt{p_i}} e^{-\beta H/2} |\sigma_i\rangle, \tag{2}$$

where $|\sigma_i\rangle$ denotes the basis of product states, $p_i = \langle \sigma_i | e^{-\beta H} | \sigma_i \rangle$, and $\beta = 1/T$ the inverse temperature. By averaging over the individual observables calculated via snapshot wavefunctions $|\psi_i\rangle$, we estimate the thermal observables,

$$
\begin{aligned}
\langle O \rangle &= \frac{1}{Z} \text{Tr}(e^{-\beta H} O) \\
&= \frac{1}{Z} \sum_i \langle \sigma_i | e^{-\beta H/2} O e^{-\beta H/2} | \sigma_i \rangle \\
&= \frac{1}{Z} \sum_i p_i \langle \psi_i | O | \psi_i \rangle,
\end{aligned}
\tag{3}
$$

where $Z$ is the partition function given by $Z = \sum_i p_i$. Hole densities and spin correlations can be calculated directly from these snapshot wavefunctions. Representative snapshots in Fig. 2 (panels (a–c) at temperatures $T = 0.200, 0.150,$ and $0.100$) reveal large hole clusters together with pronounced antiferromagnetic domains, consistent with strong hole clustering. In panel (d) at $T = 0.025$, we observe stripe correlations with intertwined charge and spin density waves. The densities of the holes at a lattice site $r$ are calculated as $n_h(r) = 1 - \langle \psi_i | n_r | \psi_i \rangle$, where $n_r = n_{r\uparrow} + n_{r\downarrow}$ and the radius of the gray circle at a site $r$ is proportional to $n_h(r)$. Spin correlations are illustrated by red and blue arrows of length proportional to $\langle \psi_i | S_0^z \cdot S_r^z | \psi_i \rangle$, where the label $r = 0$ is a reference site. Because SU(2) symmetry is intact, local

moments in a thermal mixed state are small and basis dependent; we therefore visualize correlations rather than local $\langle S^z \rangle$. To represent staggered domains, the squares denoting lattice sites are color-coded; regions with staggered correlations are shaded in one color, with adjacent staggered domains differing by a $\pi$-phase shift shown in the alternate color (either red or blue). Across all panels, we keep a fixed reference circle radius (hole density) and arrow length (spin correlator) to provide absolute visual scales. As thermal fluctuations diminish with decreasing temperature, the system increasingly favors the formation of more extensive antiferromagnetic domains (snapshots for a wider cylinder of width $W = 6$ are in the Supplementary Fig. S3). These pronounced fluctuations foreshadow the instability that ultimately condenses into stripe order. At the lower temperature $T = 0.025$ (Fig. 2d), we no longer find stochastic clustering but instead stripe order with a wavelength of ~8 lattice sites.

To capture the clustering behavior, we employ an algorithm to statistically analyze the hole clusters in the METTS snapshots. We define the set of all lattice sites in a METTS snapshot $|\psi_i\rangle$ as $\Lambda$. Each snapshot is scrutinized to identify sites where the local hole density $n_h$ exceeds a specified threshold. This threshold $n_h^{\text{th}}$ is defined as,

$$n_h^{\text{th}} = 1 - n + c\sigma_{n_h}. \tag{4}$$

where $1 - n$ is the mean hole density, $\sigma_{n_h}$ is the standard deviation of the hole density distribution within a METTS snapshot $|\psi_i\rangle$,

$$\sigma_{n_h} = \sqrt{\frac{1}{|\Lambda|}\sum_{r \in \Lambda}\left(n_h(r) - (1-n)\right)^2}, \tag{5}$$

where $|\Lambda|$ is the total number of lattice sites. Here, $c$ is a sensitivity parameter set to $c = 0.5$ to balance between detecting meaningful clusters and avoiding sensitivity to individual site variations. The sites satisfying this threshold condition are marked with black borders in METTS snapshots in Fig. 2. Our clustering analysis is robust to the threshold choice $c$ across $c \in [0.0, 0.9]$ (see Supplementary

Section IV), and while absolute cluster sizes vary, the temperature range where large clusters appear remains stable.

## Statistical analysis of clustering

We analyze a single METTS snapshot by first identifying the set of sites whose local hole density exceeds the threshold $n_h^{\text{th}}$,

$$\mathcal{E} = \left\{ r \in \Lambda \,|\, n_h(r) > n_h^{\text{th}} \right\}. \tag{6}$$

We partition $\mathcal{E}$ into disjoint clusters $\mathcal{C}$, defined as nearest-neighbor connected components of $\mathcal{E}$; two sites are connected if they share a horizontal or vertical bond,

$$\mathcal{E} = \dot{\bigcup}_{\mathcal{C}} \mathcal{C}. \tag{7}$$

For each cluster, we record its size $m = |\mathcal{C}|$ and its total hole mass,

$$\rho(\mathcal{C}) = \sum_{r \in \mathcal{C}} n_h(r). \tag{8}$$

Collecting all clusters from all snapshots, the density-weighted cluster-size probability is defined as,

$$p_m = \frac{\displaystyle\sum_{\text{snapshots}} \sum_{\substack{\mathcal{C}\,:\,|\mathcal{C}|=m}} \rho(\mathcal{C})}{\displaystyle\sum_{\text{snapshots}} \sum_{\mathcal{C}} \rho(\mathcal{C})}, \qquad \sum_m p_m = 1. \tag{9}$$

We now ask which range of hole masses supply the weight at each cluster size $m$. To do so, we coarse-grain the cluster hole mass $\rho(\mathcal{C})$ into non-overlapping windows defined by,

$$I_0 = [0, s); I_k = [k - 1 + s, k + s) \forall k = 1, 2, 3, \ldots, \tag{10}$$

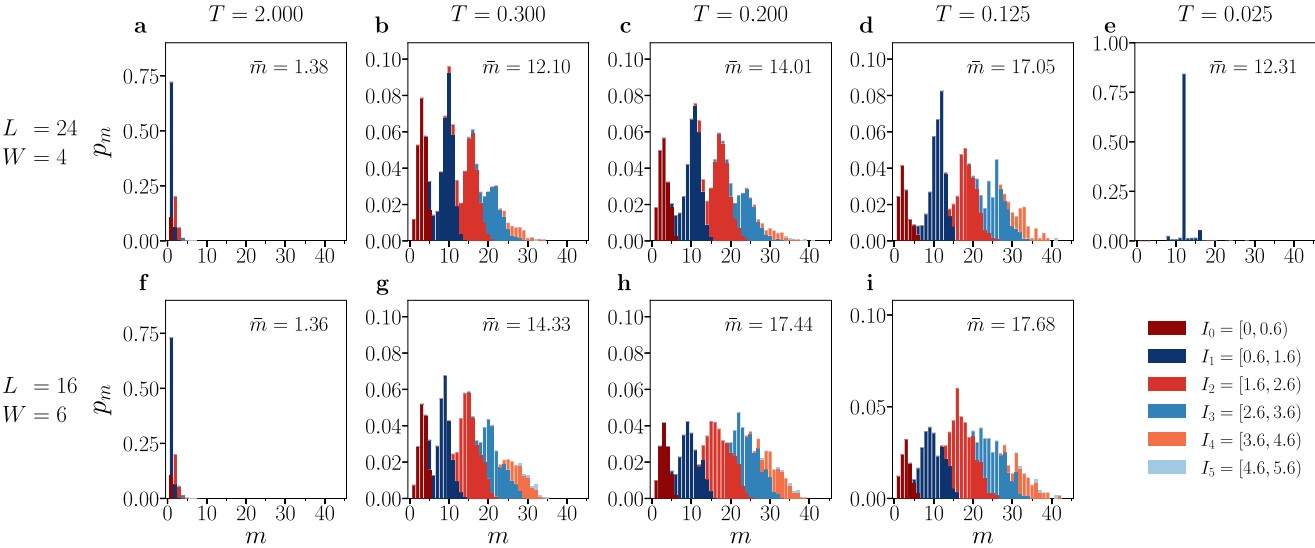

**Fig. 3 | Cluster statistics resolved by hole mass.** Stacked histograms of the cluster-size distribution $p_m$ vs. size $m$. Colors encode the cluster hole mass $\rho(\mathcal{C}) = \sum_{r \in \mathcal{C}} n_h(r)$ binned into $\mathcal{I} = \{[0, 0.6), [0.6, 1.6), [1.6, 2.6), [2.6, 3.6), [3.6, 4.6), [4.6, 5.6)\}$, so $p_m = \sum_{I \in \mathcal{I}} p_m^{(I)}$. Top row (**a**–**e**): Cylinder of size $24 \times 4$ at $T = 2.000, 0.300, 0.200, 0.125, 0.025$. Bottom row (subplots **f**–**i**): Cylinder of size $16 \times 6$ at $T = 2.000, 0.300, 0.200, 0.125$. Both geometries have $N = 96$ sites and are set to filling $n = 0.9375$. At $T = 2.0$ (subplots **a** and **f**), $p_m$ is nearly exponential, dominated by $m = 1$ $\rho(\mathcal{C}) \in [0.6, 1.6)$. On cooling, weight shifts to larger $m$ and

oscillatory lobes appear; lobe $k$ is mainly supplied by window $I_k$, consistent with AFM-assisted, near-integer hole aggregation. Because thresholding selects stripe cores ($\approx 3$ sites within an $\approx 8$-site wavelength), a stripe segment yields $\rho(\mathcal{C}) \in [0.6, 1.6)$ rather than 2. The mean cluster size $\bar{m}$ is consistently larger for $W = 6$ than for $W = 4$ at intermediate temperatures. At $T = 0.025$ (subplot **e**), $p_m$ concentrates near cluster size $m = 12$ (see Fig. 2d for a snapshot), indicating the onset of stripe order.

and define $\mathcal{I} \equiv \{I_k\}_{k \geq 0}$. We then decompose,

$$p_m = \sum_{I \in \mathcal{I}} p_m^{(I)}, \quad p_m^{(I)} = \frac{\sum_{\text{snapshots}} \sum_{\mathcal{C}: |\mathcal{C}| = m, \, \rho(\mathcal{C}) \in I} \rho(\mathcal{C})}{\sum_{\text{snapshots}} \sum_{\mathcal{C}} \rho(\mathcal{C})}. \quad (11)$$

Figure 3 shows stacked histograms of $p_m$ versus $m$, where the bar at each $m$ is built from the components $p_m^{(I)}$; thus, the total height equals $p_m$ while the color encodes the prevailing $\rho(\mathcal{C})$ window. The mean cluster size is,

$$\bar{m} = \sum_m p_m \, m. \quad (12)$$

The top panel (subplots $a - e$) presents data from a cylinder of length $L = 24$ and width $W = 4$ at $T = 2.000, 0.300, 0.200, 0.125, 0.025$. The bottom panel (subplots $f - i$) shows data from a cylinder of $L = 16$ and $W = 6$ at $T = 2.000, 0.300, 0.200, 0.125$ (no lower-$T$ data converged for this width). Both geometries have $N = 96$ sites and are set at filling $n = 0.9375$, hence the same total hole number $N_h = (1 - n)N = 6$. Therefore, the differences across panels isolate finite-width effects at fixed area. With threshold $c = 0.5$, we set $s = 0.6$ in Eq. (10) because this choice best separates the lobes of $p_m$ seen in Fig. 3; adjacent lobes differ by roughly one hole of total mass. At $T = 2.000$ the distribution is nearly exponential, dominated by $m = 1$ with $\rho(\mathcal{C}) \in I_1 = [0.6, 1.6)$. Upon cooling below $T \approx 0.75$ (see Supplementary Fig. S4 for a full temperature sweep of clustering statistics), the weight shifts to larger $m$ and the histograms develop somewhat oscillatory, peaked lobes in $m$ (roughly $1 - 5, 6 - 13, \ldots$). Successive lobes are predominantly supplied by successive mass windows $\mathcal{I}$, i.e., moving from lobe $k$ to $k + 1$ adds about one hole's worth of charge to typical clusters. Physically, the antiferromagnetic background penalizes extended domain walls; aggregating carriers in near-integer increments lets holes assemble one-by-one while minimally disturbing local AFM order, producing the observed stepwise pattern. Because thresholding isolates the high-density cores of stripes (about three sites within an ~ 8-site wavelength), a stripe segment contributes $\rho(\mathcal{C}) \in I_1 = [0.6, 1.6)$ rather than $\approx 2$. The mean cluster size $\bar{m}$ is systematically larger for $W = 6$ than for $W = 4$, demonstrating that cluster sizes grow with width. Finite-size effects are thus separable: increasing $W$ at fixed area broadens $p_m$ and raises $\bar{m}$; increasing $L$ at fixed $W$ softens the large-$m$ cutoff and reveals additional lobes (see Supplementary Fig. S13). Importantly, in the clustering window, the probability that a cluster wraps around the cylinder width remains well below unity and decreases with $W$ (see Supplementary Fig. S11). All of these are inconsistent with merely meandering stripe fluctuations at finite temperature. By contrast, below $T \approx 0.075$ (see Supplementary Fig. S4)—for example, at $T = 0.025$ as shown in subplot (e) for cylinder of size $L = 24$, $W = 4$—the distribution collapses near $m = 12$, signaling entry into the stripe regime with a sharply selected cluster size. The temperature dependence of the mean cluster size $\bar{m}$ is shown in Fig. 4a (additional dopings in Supplementary Section IV). To connect charge clustering to magnetism and to separate the clustering crossover from the onset of static stripes, we monitor the static magnetic structure factor,

$$S_{\mathrm{m}}(\mathbf{k}) = \frac{1}{N} \sum_{l,m=1}^{N} e^{i\mathbf{k}\cdot(\mathbf{r}_l - \mathbf{r}_m)} \langle S_l^z S_m^z \rangle, \quad (13)$$

evaluated at two physically motivated momenta: (i) $(\pi, \pi)$, the Néel vector that tracks antiferromagnetic correlations; and (ii) $(7\pi/8, \pi)$, the stripe ordering vector observed when stripes form on $W = 4$ cylinders at filling $n = 0.9375$[60]. Upon cooling from high $T$, $\bar{m}$ grows and develops a broad maximum within a crossover window (lavender) where $S_{\mathrm{m}}(\pi, \pi)$ is also enhanced; this reflects the formation of extended AFM domains surrounding hole clusters, i.e., the forestalled phase separation regime. At lower $T$ the magnetic weight shifts from $(\pi, \pi)$ to the

stripe vector: $S_{\mathrm{m}}(7\pi/8, \pi)$ overtakes $S_{\mathrm{m}}(\pi, \pi)$ (marked by a black dashed vertical line at $T \approx 0.06$) as static stripe correlations set in (gray shading). Concomitantly, $\bar{m}$ stops broadening and locks to the characteristic stripe cluster size, indicating that fluctuating clusters give way to a regular stripe pattern. This $(\pi, \pi) \to (7\pi/8, \pi)$ shift with decreasing $T$ is consistent with the finite-$T$ evolution reported for the same geometry in ref. 60 and with the emergence of the charge peak at $(\pi/4, 0)$ in Fig. 5. Figure 4b shows finite-size scaling of $\bar{m}(L)$ for width $W = 4$. In the clustering window ($T = 0.150$), we find sublinear power-law growth $\bar{m}(L) \sim L^\gamma$ with $\gamma \approx 0.47$, indicative of broad, system-spanning fluctuations. By contrast, at higher ($T = 2.000$) and very low temperatures ($T = 0.025$), there is almost no size dependence. For the latter, $\bar{m}(L)$ is locked at a value of around 12, characterizing the stripe cluster size.

**Charge structure factor**

Finally, we explore a more conventional approach to detecting phase separation. In Fig. 5 we present the equal-time (static) charge structure factor,

$$S_c(\mathbf{k}) = \frac{1}{N} \sum_{l,m=1}^{N} e^{i\mathbf{k}\cdot(\mathbf{r}_l - \mathbf{r}_m)} \langle (n_l - n)(n_m - n) \rangle, \quad (14)$$

for a filling of $n = 0.9375$, considering various system sizes of lengths $L = 16, 24, 32$, and $40$, with a fixed width $W = 4$. Note that, since we are working in the canonical ensemble, $S_c(\mathbf{k} = 0) = 0$, because there is no fluctuation in total density at zero momentum. At a temperature $T = 0.150$, an inner peak is present at the lowest momentum mode $\mathbf{k} = (\frac{2\pi}{L}, 0)$ resolved by the lattice, which shifts inward as the system size increases. Physically, this inward shift with system size indicates a build-up of long-wavelength density fluctuations, characteristic of PS. This trend is highlighted by dotted orange lines that mark peak positions in both the main plot and the inset. The inset further amplifies the inner peaks corresponding to different system sizes, using darker shades to represent larger sizes. Because the canonical ensemble fixes the total density, $S_c(\mathbf{k} = 0)$ must vanish, which may impose an artificial constraint at the lowest nonzero momentum, i.e. $\mathbf{k} = (2\pi/L, 0)$. However, benchmarking against free fermions and varying $U$ shows that the marked low-$k$ peak emerges only at large $U$, indicating that it reflects genuine interaction-driven density fluctuations rather than an ensemble artifact (see Supplementary Section V). As the temperature decreases to $T = 0.025$, the system transitions into a stripe order, characterized by a pronounced peak at $\mathbf{k} = (\frac{\pi}{4}, 0)$ (illustrated by a blue dashed line) that remains consistent across different system sizes.

## Discussion

We have used tensor network simulations to map the finite-temperature phase diagram of the two-dimensional Fermi-Hubbard model in the pseudogap and strange-metal regimes. At intermediate temperatures, we find pronounced hole clustering coexisting with robust antiferromagnetic (AFM) domains, signaling a strong tendency toward phase separation. Yet, upon further cooling, these clusters never coalesce into a macroscopic phase-separated state; instead, stripe order emerges. In the canonical ensemble, this "forestalled" phase separation appears as a low-momentum peak in the charge-structure factor.

AFM correlations have long been connected to pseudogap physics[43,44,60–63]. Using self-consistent constrained-path AFQMC on large periodic lattices, Xiao et al.[63] showed that the doped Hubbard model evolves from a high-$T$ disordered metallic state, nearly uniform charge and only short-range AFM fluctuations, through growing commensurate AFM correlations, into incommensurate spin-density waves, and finally a finite-$T$ stripe order where charge order is locked to the spin pattern. Our results establish an intermediate-$T$ clustering window between the disordered metal and static stripes: carriers

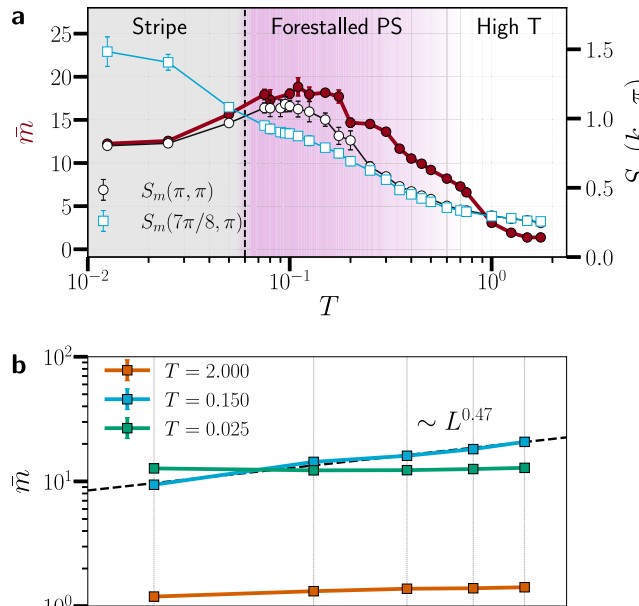

**Fig. 4 | Clustering and magnetic correlations in the finite-temperature Fermi-Hubbard model. a** Density-weighted mean cluster size $\bar{m} = \sum_m p_m m$ (maroon circles) and magnetic structure factors $S_m(\pi, \pi)$ (black open circles) and $S_m(7\pi/8, \pi)$ (blue open squares) versus temperature $T$ at $U = 10$ and $n = 0.9375$. The black dashed vertical line marks the temperature $T \approx 0.06$, where $S_m(7\pi/8, \pi)$ overtakes $S_m(\pi, \pi)$; we use this crossing as an operational definition of stripe onset. The shaded regions indicate the low-temperature stripe regime (gray) and the crossover window of forestalled phase separation (lavender), which smoothly fades into the high-temperature phase to the right. The error bars reflect $1\sigma$ uncertainty. **b** Log-log scaling of $\bar{m}(L)$ across temperatures. In the clustering regime ($T = 0.150$), $\bar{m}$ grows approximately as a power law with $L$, while at high temperature ($T = 2.000$) and deep in the stripe phase ($T = 0.025$) it remains $L$-independent.

aggregate into large, fluctuating hole-rich domains with enhanced $S_m(\pi, \pi)$ and a thermodynamic maximum in $\chi_{\text{charge}}$, yet macroscopic PS is preempted by stripe order.

Antiferromagnetic correlations are strongest when two electrons occupy neighboring sites; consequently, they favor half-filled regions and generate an effective attraction that empties neighboring areas, producing hole-rich clusters. Consistent with this, our hole-mass-resolved cluster-size histograms show stepwise aggregation: weight shifts through distinct lobes as clusters grow in near-integer (single-hole) increments at intermediate temperatures. On further cooling, the system selects a stripe order characterized by antiphase domain walls that favor pairwise rather than single-hole additions; so charge patterns settle without ever developing macroscopic phase separation. Correspondingly, the density-weighted mean cluster size increases in the clustering window and then settles to a selected value at low temperature. Resonant inelastic X-ray scattering (RIXS) experiments reveal high-temperature precursor charge density wave fluctuations and stripe correlations in cuprates[64–67], lending support to the idea that charge fluctuations play an active role in the pseudogap regime. Within AFM domains, the strong spin background opens a gap, while adjacent hole-rich metallic regions introduce low-energy states that partially fill it, reducing the density of states at the Fermi level–an archetypal pseudogap hallmark. While a detailed exploration of this mechanism is beyond our present scope, our results suggest that the interplay of AFM order and hole clustering may influence pseudogap formation. Recent high-field NMR measurements on underdoped $YBa_2Cu_3O_{6+x}$ reveal two activated spin-gap scales linked to CDW and pseudogap correlations, consistent with coexisting gapped AFM and metallic regions[68].

In the $t$-$J$ model, when $J/t \geq 1$, the ground state phase-separates[69,70]. Recent work shows that, under long-time evolution, charge degrees of freedom then remain essentially frozen[71]. This immobility may be related to the large resistivity observed in cuprates across the pseudogap, strange-metal, and bad-metal regimes[72,73].

## Methods

Solving the FHM presents substantial challenges due to its strongly interacting nature; analytical solutions are rare and limited to specific cases[74,75]. Even the most advanced numerical methods often face difficulties in accurately solving the model without biases from approximation schemes[76]. For finite temperature, the minimally entangled typical thermal states (METTS) algorithm, when applied to MPS[45,46], offers significant advantages over traditional methods (such as purification[49]), notably by representing thermal states with lower bond dimensions. METTS has facilitated insightful studies into the thermal phase diagram of the square-lattice FHM, including the study of the pseudogap regime and the stripe phase[60]. Similarly, purification techniques using iPEPS[77–80] have explored the phase diagram from high to intermediate temperatures, revealing significant distortions in antiferromagnetism upon doping[53]. Additionally, alternative approaches for finite temperature simulations using tensor networks for the Hubbard model include the exponential thermal tensor renormalization group (XTRG)[81], tangent space tensor renormalization group (tanTRG)[82], and METTS applied to PEPS[83]. A recent study using diagrammatic Monte Carlo[84] was able to achieve temperatures as low as $T = 0.067$ and up to $U = 7$ for arbitrarily large lattices, providing insights into momentum-resolved spin and charge susceptibilities. See ref. 85 for a detailed list of methods approaching the FHM at finite temperatures. Complementing these computational breakthroughs, experimental techniques with ultracold atoms have similarly advanced, enabling precise simulation and investigation of many-body physics that echo the complex interactions found in the Hubbard model[86–92]. These experiments confine systems with hundreds of fermions and can reach temperatures as low as 1/4 of the hopping energy, hosting non-trivial charge and spin correlations[93–98].

In this work, we utilize two tensor network (TN) methods to simulate the FHM, focusing on approaches that are constrained by entanglement entropy, typically characterized by the bond dimension $D$.

### Purification with Infinite Projected Entangled Pair States

The charge susceptibility calculations in Fig. 1 of the main text were obtained with the infinite projected entangled pair states (iPEPS) ansatz, using the neighborhood tensor update (NTU) algorithm[99–101], as described in ref. 53. The iPEPS operates in the thermodynamic limit, effectively eliminating finite-size effects. We focus on local updates for iPEPS optimization, called the neighborhood tensor update (NTU)[101]. This method is computationally more efficient and numerically stable compared to global updates like the Full Update. Additionally, NTU provides greater accuracy than the Simple Update[102], which does mean-field-like approximations. The iPEPS approach we used simulates thermal states using the purification technique, which is broadly favored for finite-temperature simulations. In iPEPS, the purification of a thermal state is performed using a tensor network in which the thermal density matrix $\rho(\beta)$, representing the system at inverse temperature $\beta = 1/T$, is encoded as a pure state in an enlarged Hilbert space that includes both physical and ancillary degrees of freedom (d.o.f.). The process begins by representing the infinite-temperature state as a product of maximally entangled states between corresponding local physical and ancillary d.o.f. This state serves as the starting point for imaginary-time evolution, implemented through a sequence of tensor network operations that progressively cool the system to the desired temperature. The thermal density matrix is then obtained by tracing out the ancillary degrees of freedom from the pure state

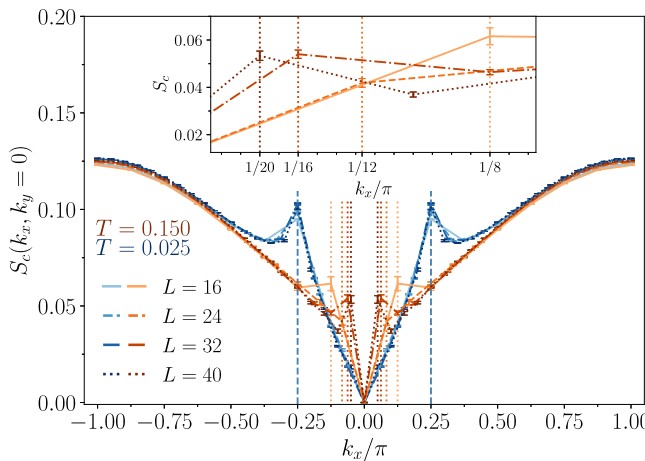

**Fig. 5 | Charge structure factor.** Static charge structure factor $S_c(\mathbf{k})$ (see Eq. (14)) for electron density $n = 0.9375$ and $U = 10$ across system sizes of length $L = 16, 24, 32, 40$ and width $W = 4$. The line style denotes system size (solid $L = 16$, dashed $L = 24$, dash-dot $L = 32$, dotted $L = 40$); color denotes temperature (blue $T = 0.025$, orange $T = 0.150$). At $T = 0.150$, a subtle inner peak appears near $\mathbf{k} = (\frac{2\pi}{L}, 0)$ and shifts inward with increasing $L$; orange dotted guides indicate $\pm 2\pi/L$. At $T = 0.025$, the onset of stripe order is marked by a robust peak around $\mathbf{k} = (\frac{\pi}{4}, 0)$ (blue vertical guides at $k_x/\pi = \pm 1/4$). The inset highlights the finite-size shifts with darker shades corresponding to larger $L$. The error bars denote $1\sigma$ standard error.

representation. Mathematically, this is described by the equation,

$$\rho(\beta) = \mathrm{Tr}_{\mathrm{anc}}\left(|\psi(\beta)\rangle\langle\psi(\beta)|\right),$$

where $|\psi(\beta)\rangle$ is the state obtained after applying the imaginary-time evolution operator $e^{-\beta H/2}$ to the initial state, and $\mathrm{Tr}_{\mathrm{anc}}$ denotes the partial trace over the ancillary degrees of freedom. In our simulations, we enforced $U(1) \times U(1)$ symmetry (which conserves particle number) using the YASTN package[103,104], optimized for fermionic and symmetric PEPS. This symmetry conservation helps to reduce computational overhead. With this approach, we could go up to bond dimensions $D = 30$ and were able to explore temperatures as low as $T = 0.125$. However, the rapid growth of the bond dimension due to the entanglement introduced by purification limits iPEPS's ability to access very low temperatures without significant computational cost.

**Minimally entangled typical thermal states with matrix product states**

The second method we employed is the minimally entangled typical thermal states (METTS) algorithm, which uses matrix product states (MPS) as the variational ansatz. METTS is especially effective in 1D cylindrical geometries and excels at lower temperatures. Unlike purification, METTS does not require the full representation of the thermal density matrix. Instead, METTS relies on generating a sequence of typical thermal states through a Monte Carlo sampling process. The method works by selecting a product state, evolving it in imaginary time by $\beta/2$, and then collapsing the evolved state into a new product state using Markov chain sampling (see Eq. (2) and Eq. (3) of the main text). The imaginary time evolution in METTS is carried out using the time-dependent variational principle (TDVP)[105,106], with subspace expansion[107], which improves the stability and accuracy of the evolution process. By avoiding the full tracking of the thermal density matrix, METTS can achieve accurate results with significantly lower bond dimensions than purification. Our METTS implementation, based on the ITensor library[108,109], was able to reach bond dimensions up to $D = 2500$, particularly for systems on $32 \times 4$ cylinder geometries, following successful approaches from previous studies[60,110–112]. A

further advantage of METTS is its ability to provide spatially resolved data, which was key in identifying the clustering at intermediate temperatures.

## Data availability
The data in the main text are made available at https://doi.org/10.6084/m9.figshare.29367464. The data for the Supplementary Information are available on request.

## Code availability
The iPEPS simulations were performed with the YASTN library[103,104]. The MPS-METTS calculations used METTS.jl (https://github.com/awietek/METTS.jl), built on top of the ITensor library[108,109]. We provide a minimal script for the Fermi-Hubbard cylinder and a README that reproduces the main figures inside the *examples* folder. The *examples* folder is the top-level directory examples/ in that repository; the script used in this work is examples/hubbard_cylinder_metts.jl.

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

## Acknowledgments

We thank Antoine Georges, George Batrouni, Salvatore Manmana, Roderich Moessner, Steve White, Luke Staszewski, Martin Ulaga, Arnab Das, Joe H. Winter, Timon Hilker, Lode Pollet, Zhenjiu Wang, Thomas Chalopin, Immanuel Bloch, Fabian Grusdt, and Annabelle Bohrdt for insightful discussions and, in particular, Fakher Assaad for highlighting the signatures of phase separation in the charge structure factor. We thank Chris Hooley for suggesting the adjective "forestalled" and for

further discussions. A.W. acknowledges support by the DFG through the Emmy Noether program (Grant No. 509755282). A.S. thanks Marek M. Rams and Jacek Dziarmaga for previous technical discussions on iPEPS. A.S. acknowledges the Alexander von Humboldt Foundation for support under the Humboldt Research Fellowship and support from the National Science Center (NCN), Poland, under project 2019/35/B/ST3/01028, at Jagiellonian University, Kraków, where part of the work was completed.

## Author contributions

A.W. conceived the project; both the authors designed research methodologies; A.S. performed the numerical simulations; both the authors contributed to the interpretation of results and writing the manuscript.

## Funding

## Competing interests

The authors declare no competing interests.
