## [Transparent Peer Review file · Nature Communications]

Forestalled Phase Separation as the Precursor to Stripe Order

Corresponding Author: Dr Aritra Sinha

Version 0:

Reviewer comments:

Reviewer #1

(Remarks to the Author)

The authors present a study of the Hubbard model with only nearest neighbor hopping, with a focus on investigate the finite temperature properties of stripes. The first finding is via iPEPS. A peak in the $q=0$ charge susceptibility is seen close to $\mu=-3$, $\langle n \rangle=0.9$. The authors then turn to the METTS method to analyze charge inhomogeneity, and demonstrate the presence of clustering in representative wavefunction snapshots. To summarize, this work uses state-of-the-art techniques to investigate and test many old ideas related to stripes and phase separation in the Hubbard model. The METTS snapshots, in particular, provide a very microscopic view of the interesting physics of the simple model, in a way that probably cannot be replicated by any other simulation short of a truly quantum simulation (using cold atoms or a quantum computer).

I think this work is important and should be published, but I have a long list of questions and suggestions for the authors.

1. In Fig. 1, it might be helpful to include as an additional plot of χ vs $\langle n \rangle$, so that one can directly see at what value of $\langle n \rangle$ is χ maximal.
2. The authors should rule out the possibility that the peak in χ vs μ is due to a van Hove singularity. The van Hove is probably not relevant at $\langle n \rangle=0.9$, but this should be supported by data, either from the same simulations or from data in literature.
3. The abstract says the peak is located around $p=1/8$. Is there a reason to say $1/8$ specifically in the abstract? Estimating the best that I can, I think the peak in Fig. 1c occurs at $\langle n \rangle = 0.91$, or $p=0.09$ rather than $1/8$. And the later METTS simulations are conducted at $p=0.0625$.
4. Why are the METTS simulations performed at $p=0.0625$ rather than at a slightly higher value?
5. In Fig. 2, why are the bottom rows of $T=0.2$ and $T=0.15$ covered up?
6. In Fig. 5 and the discussion of the peak in the structure factor at the smallest nonzero momentum point, I have a more mundane explanation. (*) First, because the canonical ensemble is used, there is no fluctuation of the total density, the structure factor at $q=0$ has to vanish. (**) On the other hand, if we think about the local correlations, which correspond to the sum of the structure factor over all q , they should not be affected by the choice of ensemble. Could it be that for a grand canonical ensemble calculation, the structure factor is smooth through $q=0$, but when the canonical ensemble is used, a peak develops at the smallest nonzero q in order to satisfy both (*) and (**)?

One way to test this is to simulate a non-interacting model in the canonical ensemble with open boundary conditions, and check if something similar happens in the structure factor.

I am in favor of publishing this manuscript in Nature Communications if the authors can satisfactorily address these points.

(Remarks on code availability)

Reviewer #2

(Remarks to the Author)

The manuscript under review investigates the behavior of a slightly doped Hubbard model on a square lattice, focusing on the transition from a disordered state dominated by high-temperature thermal fluctuations to the formation of long-range

stripe order at low temperatures. The study is conducted using iPEPS and METTS methods, and the main conclusion is that an incipient charge clustering promoted by antiferromagnetic correlations at intermediate temperatures, suggesting near phase separation, which is ultimately preempted by stripe order at lower temperatures.

While the results presented are interesting, it is worth noting that similar findings have already been reported in a previous study, *Physical Review X* 13, 011007 (2023). In that work, the system also exhibited a disorder phase at high temperatures due to thermal fluctuations, and at low temperatures, a coexistence of SDW and CDW stripe phases was observed. Moreover, the authors pointed out the appearance of short-range magnetic order prior to the onset of stripe order at intermediate temperatures. Given that the ground state of the system in both works exhibits long-range stripe order, the mechanisms behind the formation of these short-range ordered structures are likely to be consistent across both studies.

Considering the nature of the findings and the fact that similar results have already been published, the manuscript does not appear to offer a sufficiently novel contribution to warrant publication in *Nature Communications*. The work would likely be better suited for submission to *Communications Physics*.

(Remarks on code availability)

Reviewer #3

(Remarks to the Author)

Thank you for forwarding the paper by Sinha et al. on Forestalled Phase Separation as the Precursor to Stripe Order.

The paper represents an interesting novel twist on the standard one-band 2D Hubbard model at intermediate temperatures suggesting phase separation. The paper uses large scale state-of-the-art numerical simulations for the study which are well-controlled, and in this sense, considered numerically exact. The paper is well written, and the analysis is (mostly) well polished.

Hence I recommend publication once the following points have been addressed still.

Main criticisms:

The size of the characteristic stripe cluster in Fig. 3b appears to show up as a natural unit for the location of the peaks in the lower panels (Fig. 3cd). The paper does not appear to comment on this at all. Yet this seems to suggest that the phase separation already also knows about the stripe order! Meaning that the typical cluster sizes are (near) commensurate with the strip sizes. Can this really be seen as an 'unbiased' phase separation then as suggested by the title of the paper?

The paper comments on the phase transition at finite t' yet chooses to set $t'=0$ and analyze finite doping. It would be very interesting to see if the clustering also persists when turning on t' with no doping, instead.

Further minor comments:

The presentation of the clustering needs be more concise:

- what is the definition of 'maximally connected component of sites' (connectivity to at least one site horizontally or vertically?)

- C_l specifies the cluster indexed by l . The cluster size then is given by $|C_l|$ (mentioned once as such). But then cluster size is just specified as C_l in the figures which is a rather sloppy notation. This should be $|C|$ (without index l because it's irrelevant), or more verbose like 'cluster size', etc.

- the terminology 'normalized frequency' is non-standard: the frequency of cluster sizes in Eq. (8) is really just a plain normalized probability. Yes, it may be translated into a rate of occurrence,

but Eq. (8) is dimensionless. Hence in what follows, let me refer to Eq. (8) as probability p_n to find cluster size n .

- the probability p_n diminishes with increasing n ; hence to increase readability (?), Fig. 3 plots $n \cdot p_n$. Yet the overall sum then already also represents the mean cluster size \bar{n} . A sentence along these lines may be helpful.

- given the $\sim \sqrt{L}$ scaling of \bar{n} in Fig. 4b, is this already also reflected in the tail distributions in Fig. 3? What is the power-law seen in the tail?

The 'structure factors' for spin and charge are always the 'static structure factors', i.e., with no dynamics.

(Remarks on code availability)

Version 1:

Reviewer comments:

Reviewer #1

(Remarks to the Author)

The authors have gone above and beyond in addressing all my concerns from the previous round of review. I am pleased to say that I have no further comments. I think this manuscript is a very valuable contribution to the literature of Hubbard model physics and I endorse its publication.

(Remarks on code availability)

It would be great if the METTS repository contained a quick guide to using the code. Currently there is no such documentation.

Reviewer #3

(Remarks to the Author)

Thank you for forwarding the revised paper by Sinha et al. on Forestalled Phase Separation as the Precursor to Stripe Order.

The authors included many of the suggestions in their revision which I appreciate. My main criticism, however, persists if not strengthened. Hence I am afraid, the clustering analysis needs further improvement before I can make a recommendation.

My major worries remain finite-size effects in the clustering analysis, and further questions on the data itself.

One may look at the present analysis coming from low-T: starting from stripe order, stripes start to move. As such they will get closer to each other at some places but move away from each other at other places. Based on an arbitrary but fixed algorithm to determine cluster sizes, this also will naturally lead to an increase in cluster sizes. Specifically, in the presence of finite width- W cylinders, this also would explain the pronounced commensurate peaks currently seen in the $m \cdot p_m$ distributions, as well as the clustering wrapping around the full circumference in the snapshots such as Fig. 2. This weakens the argument on forestalled phase separation.

The wider spread in the distributions $m \cdot p_m$ for intermediate temperatures partly may also be seen as an artifact of the presentation of the data that is biased towards larger m

by plotting $p_m \rightarrow m \cdot p_m$.

Lastly, it is known that there is a π phase-shift across AF domains (stripes) which is only briefly mentioned in the SM, but not much addressed otherwise. Yet is it possible that due to finite-T two stripes can close the space in between them when moving closer to each other, potentially with a weak attraction due to better satisfied nearest-neighbor AF correlations?

In more detail:

Fig. 3: the data in the newly added insets to panels (c-d) bends downwards on a log-log scale. Hence I see no reason why to fit a power law there, instead of the exponential same as in panel (a). Then with $p_m \sim \exp(-(m/(W \cdot \xi)))$ with correlation length ξ along the cylinder of width $W=4$, the data $m \cdot p_m$ in the main panels of Fig. 3 is peaked at $m_{pk} = W \cdot \xi$, suggesting $\xi \sim m_{pk}/W < \sim 5 = O(W)$. Then one may phrase the data in Fig. 3 as follows: the correlation length increases up to order of the width of the cylinder before it decreases again.

Hence there are strong finite-size effects related to the chosen cylinder of width W on top of the period $m \sim (8,12) \sim (2,3)W$ that appears as a single peak in the $T \rightarrow 0$ data at $m=3W$. This also appears to hold for the $W=6$ cylinder in the SM where breaking of the stripes across the circumference may be due to a difficulty converging the thermal state.

Fig. 2: I miss absolute values in this plot: there should be a legend in terms of a particular circle that indicates a particular hole concentration $p_i = 1 - \langle n_i \rangle$. Currently, the variations seen in the hole concentration in Figs. 2 (MP) and 3, 4, 9 are only on a relative basis. I may have missed it, but I also did not see any values specified for the standard deviation σ [Eq. 4] (besides that σ is also used as spin index).

Ditto for the spin expectation values: the value corresponding to a particular length of the arrow should be given as a quantitative reference.

Why are the hole distribution vs spin distribution shown on an inequivalent footing?

The hole concentrations are specified as local expectation values, whereas spin expectation values which are specified as (long-range) static correlations [black cross in Figs. 2 (MP), 3, 4, and 9 (SM) at rather arbitrary locations picked inside a spin cluster]. Is it because the plain local expectation values for the spin are very small because of the thermal mixed state? If so, this deserves a brief discussion / a few comments on its own.

There are surprisingly many sizable parallel spins in Fig. 2 in the hole clusters for the lowest temperature $T=0.1$ (panel c) (for a completely depleted site there would be no spin correlations).

Somehow the cluster sizes in the snap shot Fig. 2a does not seem to exceed $m < \sim 12 = 3W$, whereas the respective distribution in Fig. 3c (also at $T=0.20$) suggests much larger clusters to be present. In this sense, is Fig. 2a representative (are the T-labels correct?), or does Fig. 3c overemphasize the tail by plotting $p_m \rightarrow m \cdot p_m$?

Looking at Fig. 2a assuming [$T \rightarrow 0$ (!)]: there seems to be

a horizontal period of $(2+6=8)$. With a doping of $p=1/16$
[$n=0.9375 = (1-1/16)$; why not write $p=1/16$ instead of $n=0.9375$?]
this suggests a half-filled stripe. This actually may prevent
two stripes from completely merging with each other
while fully satisfying AF order?

Very minor remarks:

I suggest to order the panel labels in the order of T.

Please add labels

- a,b,c to Fig. 2

- a-d to Fig. SM-9

Fig. SM-6: what is the doping in panel (a),
and the constant c in panel (b)?

(Remarks on code availability)

Version 2:

Reviewer comments:

Reviewer #3

(Remarks to the Author)

Thank you for forwarding the second revision of the paper
by Sinha et al. on Forestalled Phase Separation as the
Precursor to Stripe Order.

The authors put significantly more effort still into their
analysis. The revised figures work well now. In particular
I like the newly added color coding w.r.t. the average
integer hole mass. A central concern of mine was that stripes
wrap around the cylinder and thus linke to finite size effects.
This is now also addressed in detail in the paper and much
emphasizes the original claim of the paper. With these
updates in place I recommend publication in Nature Communications.

Final recommendations:

The doping of $1/16$ suggests a minimal cluster size of
16 sites to localize one dopant hole.
Assuming directional invariance, such a cluster may thus
be concentrated onto a circle of diameter $d \sim 4.5$.
Now if the width of a cylinder is smaller than this,
most clusters will actually wrap the cylinder which
will manifest as 'stripes'.
E.g. this holds for the $W=4$ cylinder in Figs. 2 or S8.
On the other hand, when $W>d$ as for $W=6$ in Fig. 8
or the lower panels in Fig. 3, the wrapping of the
'stripes' around the cylinder will be considerably
broken up into puddles ('clusters') as seen in Fig. S3.
This suggests that for a larger doping of $1/8$ with $d \sim 3.19$,
the breaking up of the stripes around the cylinder
already should also be considerable for $W=4$. It would
be very instructive to see this picture supported,
specifically so given that there is much literature
on the Hubbard model for doping $1/8$. Based on Fig. S5,
apparently such data already exists.

Fig. 2d shows 4 stripes which according to color coding
in Fig. 3e should carry a hole mass of about 1 (1h).
However, the system in Fig. 2 has a doping of $1/16$
and hence carries a total of $8 = 32 \cdot 1/16$ holes.
Where does the difference of 4 holes go?
Being close to the ground state, Fig. 2d rather

suggests 2h per stripe. Is this discrepancy an artifact of the choice of $c=0.5$? If yes, then isn't the choice of the intervals I_n , and hence the color coding in Fig. 3 strongly sensitive on the particular choice of c ? Did the choice of $c=0.5$ happen to be an accidentally good choice in this regard? Please clarify.

(Remarks on code availability)

Reply to referees

Forestalled Phase Separation as the Precursor to Stripe Order

Aritra Sinha, Alexander Wietek

Nature Communications submission NCOMMS-25-09251 Sinha

1. Summary of changes

We have substantially revised the manuscript in response to the three referees' comments. In the rest of our reply we will refer to the main text as **MT** and the Supplementary Material as **SM**. The key updates include:

- **Abstract update:** removed the imprecise “ $p = 1/8$ ” in the abstract.
- **Fig. 1(e):** added a new panel plotting χ_{charge} vs. density n to pinpoint the maximum at around filling $n \approx 0.91$.
- **Van Hove origin possibility:** included a comparison to noninteracting DOS and plaquette CDMFT results to rule out a band-structure origin for the charge susceptibility peak using iPEPS.
- **MT Fig. 2 cleanup:** removed the stylistic overlays that masked the bottom rows at $T/t = 0.2$ and 0.15 , ensuring all METTS snapshots are fully visible.
- **Supplementary Fig. 5 (SM Sec. II):** added a new figure compiling $p_m m$ histograms across $T = 2.0 \rightarrow 0.0125$, showing the evolution from exponential suppression to algebraic tails to a $m = 12$ peak.
- **Histogram consistency:** because our focus is on hole clustering, cluster histograms must reflect the fraction of *holes* in size- m clusters. In the original submission, the histograms in Fig. 3 used a simple count based definition of probability, while Fig. 4 and the SM relied on the density-weighted probability p_m (defined in Eq. 8 of the main text) and its mean (Eq. 9). We have now standardized on this density-weighted p_m throughout the text. We also changed the temperature being inspected at MT Fig. 3(d) from $T = 0.1$ to $T = 0.075$ to choose a broader spread of the selected temperatures.
- **Ensemble-artifact tests (SM Sec. IV, Figs. 7–8):** added free-fermion grand-canonical vs. canonical structure factors and an interacting U -scan, proving the small- q peak is a strong correlation effect, not an ensemble artifact.
- **Finite- t' study (SM Sec. V, Figs. 9–10):** presented METTS snapshots at a particular value of $t'/t = 0.30$ for both half-filling ($n = 1.00$) and light doping ($n = 0.9375$), showing clustering requires mobile holes and does not arise from t' alone.
- **Cluster definition and notation:** clarified that clusters are nearest neighbor connected components (MT Eq. 7), replaced all C_l with the scalar $m = |C|$, and unified the notation throughout.
- **Probability formalism:** removed the term “normalized frequency,” in MT Eq. (8) to define the probability p_m , and consistently use p_m in text and figures.

- **Fig. 3 insets:** we try to fit a power law with the raw probabilities p_m within the fitting windows ($12 < m < 48$) and report the fitted exponents $\alpha \approx 2.89$ ($T = 0.20$) and $\alpha \approx 2.81$ ($T = 0.075$).
- **Discussion of Xiao et al. PRX 2023:** we have described the important reference in the Discussion section and highlighted its similarities and differences from our work.

2. Reply to the First Referee

The authors present a study of the Hubbard model with only nearest neighbor hopping, with a focus on investigate the finite temperature properties of stripes. The first finding is via iPEPS. A peak in the $q=0$ charge susceptibility is seen close to $\mu=-3$, $\langle n \rangle=0.9$. The authors then turn to the METTS method to analyze charge inhomogeneity, and demonstrate the presence of clustering in representative wavefunction snapshots. To summarize, this work uses state-of-the-art techniques to investigate and test many old ideas related to stripes and phase separation in the Hubbard model. The METTS snapshots, in particular, provide a very microscopic view of the interesting physics of the simple model, in a way that probably cannot be replicated by any other simulation short of a truly quantum simulation (using cold atoms or a quantum computer).

I think this work is important and should be published, but I have a long list of questions and suggestions for the authors.

We thank the referee for these encouraging remarks and for the recommendation to publish.

1. In Fig. 1, it might be helpful to include as an additional plot of χ vs $\langle n \rangle$, so that one can directly see at what value of $\langle n \rangle$ is χ maximal.

We have added exactly that extra subplot: in the revised Fig. 1 e, χ_{charge} vs n is plotted and clearly shows the maximum occurring around $n \approx 0.91$.

2. The authors should rule out the possibility that the peak in χ vs μ is due to a van Hove singularity. The van Hove is probably not relevant at $\langle n \rangle=0.9$, but this should be supported by data, either from the same simulations or from data in literature.

We now explicitly discuss why the observed χ_{charge} peak at $n \approx 0.91$ is not a van Hove effect. On MT page 2 and 3, lines 127 – 137, we cite plaquette CDMFT results showing that any noninteracting van Hove DOS divergence would occur at much larger doping (filling n roughly closer to 1) for comparable U . In particular we write:

Importantly, plaquette CDMFT studies of the square-lattice Hubbard model at strong coupling [reymbaut2019] show that the van Hove singularity (defined via the maximum in the local density of states at the Fermi level) appears at much larger doping than our observed susceptibility peak near $n \approx 0.91$ for comparable Coulomb strength U . This strongly suggests that our result does not arise from a noninteracting band-structure effect, but instead reflects genuine strong-coupling physics associated with Mottness.

Thus, we have ruled out a simple van Hove explanation.

3. The abstract says the peak is located around $p=1/8$. Is there a reason to say $1/8$ specifically in the abstract? Estimating the best that I can, I think the peak in Fig. 1c occurs at $\langle n \rangle = 0.91$, or $p=0.09$ rather than $1/8$. And the later METTS simulations are conducted at $p=0.0625$.

We have corrected the abstract. It now reads (MT page 1, line 13):

This maximum is located around filling $n = 0.91$...

The reference to $1/8$ doping has been completely removed.

4. Why are the METTS simulations performed at $p=0.0625$ rather than at a slightly higher value?

To demonstrate that our conclusions do not hinge on this particular filling, SM Fig. 6(b) tracks the mean cluster size for five dopings from $p = 0.047$ to $p = 0.156$. All exhibit the same broad maximum near $T \sim 0.1$, illustrating that clustering is robust over a broad range in hole density. We retained $p = 0.0625$ in the MT figures solely because convergence with bond dimension was easier for this doping. This allowed us to perform simulations even for a larger width cylinder $W = 6$; we show those results in SM Fig. 3. We add a comment on this at MT page 3 lines 166 – 169.

5. In Fig. 2, why are the bottom rows of $T=0.2$ and $T=0.15$ covered up?

The earlier overlay was intended purely as a stylistic emphasis; to avoid any ambiguity we have made all the lattices fully visible in the revised Fig. 2.

6. In Fig. 5 and the discussion of the peak in the structure factor at the smallest nonzero momentum point, I have a more mundane explanation. (*) First, because the canonical ensemble is used, there is no fluctuation of the total density, the structure factor at $q=0$ has to vanish. (**) On the other hand, if we think about the local correlations, which correspond to the sum of the structure factor over all q , they should not be affected by the choice of ensemble. Could it be that for a grand canonical ensemble calculation, the structure factor is smooth through $q=0$, but when the canonical ensemble is used, a peak develops at the smallest nonzero q in order to satisfy both (*) and (**)?

One way to test this is to simulate a non-interacting model in the canonical ensemble with open boundary conditions, and check if something similar happens in the structure factor.

We thank the reviewer for this important point. To address it, SM Sec. IV now presents a direct comparison between free-fermion and interacting METTS structure factors on the same 32×4 cylinder at $T = 0.15$:

- Free-fermion, grand-canonical (analytic): $S_c(k)$ is perfectly smooth through $k = 0$ and $k_{\min} = 2\pi/L$.
- Free-fermion, canonical (snapshot Monte Carlo): enforcing fixed N drives only a tiny, order- $1/N$ redistribution into k_{\min} .

We show this in Fig. 1 here (and SM Fig. 7) and compare it with $U = 10$ METTS results which highlight the great difference between the strong coupling and free fermion scenarios and cannot be ignored as an ensemble effect.

Figure 1: **Charge Structure Factor — METTS vs free fermion comparisons.** Charge-structure factor at $n = 0.9375$ for a cylindrical lattice of length $L = 32$ and width $W = 4$. **Left:** analytic grand-canonical result **Centre:** canonical result from free fermion snapshots. **Right:** interacting $U = 10$ METTS. Only the interacting case exhibits the subtle inner peak at $k_x = 2\pi/L$ for intermediate temperatures $T = 0.3, 0.15$ and a sharp peak at $k_x = \pi/4$ for the charge-density-wave at $T = 0.025$. The presence of the inner peak at intermediate temperatures uniquely in the interacting ($U = 10$) canonical METTS data suggests genuine emergent long-wavelength density correlations rather than numerical artifacts or trivial ensemble effects.

A complementary U -scan (Fig. 2 here and SM Fig. 8) further shows that this inner peak remains negligible for $U \lesssim 6$ and grows sharply only in the strong-coupling regime. These results demonstrate that while the canonical constraint mathematically enforces $S_c(0) = 0$, it cannot by itself produce the substantial k_{\min} enhancement we observe; only genuine interaction-driven correlations can.

We have added a concise summary of these findings to the MT page 5 lines 301 – 309 to clarify that the small- k peak in MT Fig. 5 is a hallmark of incipient charge clustering, not a trivial ensemble effect. As we do not see full phase separation in our snapshots, we do not expect this peak to diverge, but we still feel it is important to report this observation.

I am in favor of publishing this manuscript in Nature Communications if the authors can satisfactorily address these points.

We thank the referee again for constructive comments which helped to significantly improve the quality of our paper and once again appreciate their continued recommendation for publication.

Figure 2: **Charge Structure Factor for different Coulomb Strengths.** Charge structure factor $S_c(k_x, k_y=0)$ for an $L \times W = 16 \times 4$ cylinder at $T = 0.15$ obtained with METTS in the canonical ensemble. The non-interacting system ($U = 0$) shows no inner peak; only at strong coupling ($U = 10$) does a clear albeit tiny peak emerge at the smallest non-zero momentum $k_x = \pm 2\pi/L$. The emergence of a small peak exclusively at strong coupling ($U = 10$) unambiguously connects observed density correlations to interaction-driven physics, rather than trivial finite-size or ensemble effects.

3. Reply to the Second Referee

The manuscript under review investigates the behavior of a slightly doped Hubbard model on a square lattice, focusing on the transition from a disordered state dominated by high-temperature thermal fluctuations to the formation of long-range stripe order at low temperatures. The study is conducted using iPEPS and METTS methods, and the main conclusion is that an incipient charge clustering promoted by antiferromagnetic correlations at intermediate temperatures, suggesting near phase separation, which is ultimately preempted by stripe order at lower temperatures.

While the results presented are interesting, it is worth noting that similar findings have already been reported in a previous study, *Physical Review X* 13, 011007 (2023). In that work, the system also exhibited a disorder phase at high temperatures due to thermal fluctuations, and at low temperatures, a coexistence of SDW and CDW stripe phases was observed. Moreover, the authors pointed out the appearance of short-range magnetic order prior to the onset of stripe order at intermediate temperatures. Given that the ground state of the system in both works exhibits long-range stripe order, the mechanisms behind the formation of these short-range ordered structures are likely to be consistent across both studies.

Considering the nature of the findings and the fact that similar results have already been published, the manuscript does not appear to offer a sufficiently novel contribution to warrant publication in *Nature Communications*. The work would likely be better suited for submission to *Communications Physics*.

We thank the referee for deeming our results interesting and bringing this important reference to our attention, which we have cited in the first submission but not explicitly discussed. After repeated careful reading of the mentioned paper, we are unable to arrive at the referee's conclusion that "similar findings have already been reported in a previous study". The central result of our manuscript is the clustering of charge carriers (forestalled phase separation) in a regime above the transition to the stripe order. In no way is such an observation mentioned in the paper mentioned by the referee. While both studies are concerned with the evolution of spin and charge degrees of freedom of the Hubbard model upon cooling and arrive at similar

conclusions about the onset of stripe order, the novel, intriguing discovery of clustering above the transition temperature is unique to our manuscript. We would also like to emphasize that this is not a minor add-on of our work, but in fact the central statement, as apparent in our title, abstract and discussion.

However, we very much appreciate the referee bringing the results of Xiao et al. to our attention, and would like to use this as an opportunity to spell out how precisely our study differs from Xiao et al. (PRX 13, 011007 (2023)). Xiao et al. indeed describe a high-T disordered regime that flows, through short-range antiferromagnetic correlations, into a low-T state where spin-density-wave and charge-density-wave stripes coexist. The study is performed using the constrained-path auxiliary field quantum Monte Carlo approximation, which has had numerous successes in the past and often exhibits close agreement with low-temperature DMRG methods. The evidence for the occurrence of the stripe state is obtained by applying pinning fields, which slightly distort the system to favor finite order parameters in finite systems. Using these approximations, the authors present impressive results on large square geometries.

It should be noted that checking an approximate method with an unbiased method, such as METTS, in the present manuscript, adds further confidence in the obtained results. And indeed, on the subset of phenomena both our studies focus on, we find close agreement and arrive at the same physical conclusions.

We do, however, provide novel analyses and insights that go beyond the analysis performed by Xiao et al. We would like to be explicit about this:

1. We provide accurate iPEPS data in the thermodynamic limit on the charge susceptibility χ_{charge} . This data is a central piece of our argumentation for clustering, and is not provided by Xiao et al.. Moreover, we also consider this a significant technical achievement.
2. Consequently, the occurrence of a peak in χ_{charge} has not been detected by Xiao et al.
3. We analyse METTS snapshots and perform detailed statistics of cluster formation. This analysis is an entirely novel approach; consequently, no such analysis has been performed by Xiao et al.
4. The conclusion of this analysis is the occurrence of clustering, a physical phenomenon not discussed in Xiao et al.
5. Our simulation using METTS is performed in the canonical ensemble, which is an alternative to grand canonical simulations as in Xiao et al., which could be of crucial importance when detecting phase separation.
6. Our simulation is performed using an unbiased method. While this prevents us from simulating the impressive system sizes of Xiao et al., it strongly supports their physical picture by confirming their observations on smaller lattices.
7. The occurrence of a clustering regime at intermediate temperatures has eluded the analysis in Xiao et al. We do think this discovery is of vital importance to understanding the pseudogap and strange metal regimes in the two dimensional Hubbard model.

To summarize, both our studies are concerned with the same physical model. However, our methods and novel analyses reveal information which go significantly beyond the insights obtained in Xiao et al.. While we confirm the subset of findings in Xiao et al. using an unbiased numerical method, the key observation of clustering is entirely unique to our manuscript. Now we added a few comments about Xiao et al. reference in the **Discussion** section of MT:

Using self-consistent constrained-path AFQMC on large periodic lattices, Xiao et al. [**xiao2023**] showed that the doped Hubbard model evolves from a high- T disordered metallic state, nearly uniform charge and only short-range AFM fluctuations, through growing commensurate AFM correlations, into incommensurate spin-density waves, and finally a finite- T stripe order where charge order is locked to the spin pattern. Our tensor-network study uncovers an additional intermediate window between the disordered metal and static stripe order: here, charge carriers aggregate into fluctuating clusters and produce a pronounced peak in χ_{charge} , signaling an incipient but ultimately forestalled phase separation.

4. Reply to the Third Referee

Thank you for forwarding the paper by Sinha et al. on Forestalled Phase Separation as the Precursor to Stripe Order.

The paper represents an interesting novel twist on the standard one-band 2D Hubbard model at intermediate temperatures suggesting phase separation. The paper uses large scale state-of-the-art numerical simulations for the study which are well-controlled, and in this sense, considered numerically exact. The paper is well written, and the analysis is (mostly) well polished.

Hence I recommend publication once the following points have been addressed still.

We thank the referee for finding our paper interesting and recommending it for publication.

Main criticisms:

The size of the characteristic stripe cluster in Fig. 3b appears to show up as a natural unit for the location of the peaks in the lower panels (Fig. 3cd). The paper does not appear to comment on this at all. Yet this seems to suggest that the phase separation already also knows about the stripe order! Meaning that the typical cluster sizes are (near) commensurate with the strip sizes. Can this really be seen as an ‘unbiased’ phase separation then as suggested by the title of the paper?

We thank the referee for this insightful point. First, we apologize for the confusion in our earlier version, where we inadvertently plotted a count-based histogram instead of the density-weighted probability distribution p_m , we regard the latter as more physically meaningful. To address this fully and illustrate clustering across all regimes, Section II of the SM now includes Fig. 5 (here Fig. 3), which shows $p_m m$ versus m for temperatures ranging from $T = 2.0$ down to $T = 0.0125$. These histograms reveal that, although clusters of size $m = 12$ are prominent at every temperature, they only become the single largest contributor below $T \approx 0.10$, and producing the broad maximum seen in MT Fig. 3(d) at $T = 0.075$. Crucially, our METTS snapshots are generated under the unbiased Hubbard Hamiltonian with no pinning fields or predefined wavelengths; so the spontaneous predominance of $m = 12$ is part of same short-range antiferromagnetic interaction physics that later stabilize the stripe order, while the continued high- m tail confirms a genuinely clustering-dominated, ‘unbiased’ forestalled phase-separation regime rather than a preformed stripe phase. And even though the clusters of size 12 can be prominent, this does not necessarily mean stripe order with a regular CDW pattern as the size 12 clusters can still fluctuate.

The paper comments on the phase transition at finite t' yet chooses to set $t'=0$ and analyze finite doping. It would be very interesting to see if the clustering also persists when turning on t' with no doping, instead.

We thank the referee for raising this important point and we devote Sec. V of the SM to address this. To test the role of next-nearest hopping in driving clustering, we carried out METTS simulations at $t'/t = 0.30$ on a 32×4 cylinder at $T = 0.15$ for both half-filling ($n = 1.00$) and light doping ($n = 0.9375$). SM Fig. 9 presents representative METTS snapshots: at half-filling the density remains essentially uniform (no hole-rich regions), whereas at $n = 0.9375$ distinct hole clusters are clearly visible against large antiferromagnetic domains. The corresponding cluster-size probability histograms in SM Fig. 10 show an exponential decay at $n = 1.00$ but recover the broad, algebraic tails characteristic of the $t' = 0$ case once holes are introduced. These results confirm that t' by itself does not induce clustering; the phase-separation-like behavior we report arises only in the presence of hole doping. A comprehensive mapping of the t' -doping phase diagram

Figure 3: Evolution of cluster-size distributions in METTS snapshots across temperatures. The panels plot $p_m m$ versus cluster size m for a 32×4 cylinder at $U = 10$, $n = 0.9375$, with temperatures labeled in each subplot. This comprehensive temperature evolution plot broadly distinguishes between the three characteristic regimes discussed in the main text: disordered (high T), cluster-rich (intermediate T), and stripe-ordered (low T). At high temperatures ($T = 2.0, 1.5, 1.0$) large clusters are exponentially suppressed; at intermediate temperatures ($T = 0.5\text{--}0.075$) the distributions broaden with tails consistent with algebraic decay; and at the lowest temperatures ($T = 0.05, 0.025, 0.0125$) a pronounced peak at $m = 12$ emerges, marking the onset of striped clusters.

would be an interesting direction for future work, but the present data demonstrate that finite- T clustering is a genuine hole-driven effect, not an artifact of perfect nesting.

Further minor comments:

The presentation of the clustering needs be more concise: - what is the definition of ‘maximally connected component of sites’ (connectivity to at least one site horizontally or vertically?)

Thank you for pointing out this omission. We have added the following sentence to the MT page 3, lines 226 – 228):

We partition \mathcal{E} into disjoint *clusters* \mathcal{C} , defined as nearest-neighbor connected components of \mathcal{E} ; two sites are connected if they share a horizontal or vertical bond ...

This precisely fixes the connectivity rule.

- \mathcal{C}_l specifies the cluster indexed by l . The cluster size then is given by $|\mathcal{C}_l|$ (mentioned once as such). But then cluster size is just specified as \mathcal{C}_l in the figures which is a rather sloppy notation. This should be $|\mathcal{C}|$ (without index l because it’s irrelevant), or more verbose like ‘cluster size’, etc.

We thank the reviewer for highlighting this notational inconsistency. In the revised manuscript every occurrence of \mathcal{C}_l has been replaced by the simple $m = |\mathcal{C}|$ which unambiguously denotes cluster size. Equations (8)–(9), in the MT, and all figure legends now consistently refer to cluster size m . This change removes any ambiguity about indexing.

- the terminology ‘normalized frequency’ is non-standard: the frequency of cluster sizes in Eq. (8) is really just a plain normalized probability. Yes, it may be translated into a rate of occurrence, but Eq. (8) is dimensionless. Hence in what follows, let me refer to Eq. (8) as probability p_n to find cluster size n .

We have removed “normalized frequency” throughout. Eq. 8 in the MT now reads

$$p_m = \frac{\sum_{\text{snaps}} \sum_{\mathcal{C}:|\mathcal{C}|=m} \rho(\mathcal{C})}{\sum_{\text{snaps}} \sum_{\mathcal{C}} \rho(\mathcal{C})}, \quad \sum_m p_m = 1$$

where the total hole density of a cluster is $\rho(\mathcal{C}) = \sum_{i \in \mathcal{C}} n_h(i)$, and p_m is introduced everywhere as the probability that a hole belongs to a cluster of size m .

- the probability p_n diminishes with increasing n ; hence to increase readability (?), Fig. 3 plots $n * p_n$. Yet the overall sum then already also represents the mean cluster size \bar{n} . A sentence along these lines may be helpful.

We agree that spelling out the connection between the plotted histogram and the mean cluster size makes the presentation clearer. Accordingly, we have added the following sentence clarifying sentence at the end of the caption of MT page 4 lines 233 – 236:

Figure 3 plots $p_m m$ versus m , which highlights the contribution of each cluster size to the average. Since

$$\bar{m} = \sum_m m p_m, \quad (9)$$

the area under the $p_m m$ curve in each panel equals the mean density-weighted cluster size.

- given the \sqrt{L} scaling of \bar{m} in Fig. 4b, is this already also reflected in the tail distributions in Fig. 3? What is the power-law seen in the tail?

We thank the reviewer for this insightful question. In the revised manuscript, the insets of MT Fig. 3(c) and (d) display tails consistent with algebraic decay, $p_m \propto m^{-\alpha}$, fitted over $12 < m < 48$, with exponents $\alpha \approx 2.89$ at $T/t = 0.20$ and $\alpha \approx 2.81$ at $T/t = 0.075$. We have also added an inset to MT Fig. 3(a) at the high temperature $T = 2.0$, which follows a clear exponential decay. While these fits suggest the possible presence of power-law behavior, the limited system sizes prevent a precise determination of how the tail exponent alone governs the $\bar{m} \sim \sqrt{L}$ scaling seen in MT Fig. 4(b). We therefore defer a quantitative connection between the tail exponent and system-size scaling to future, larger-scale studies.

The ‘structure factors’ for spin and charge are always the ‘static structure factors’, i.e., with no dynamics.

We thank the reviewer for this important clarification. Indeed, all of our spin and charge correlators are computed at equal time and carry no frequency dependence. To make this explicit, we have inserted the word “static” while defining the magnetic and charge structure factors in MT.

We thank our third referee again for the interesting questions and suggestions which we believe led to significant improvement in our work and we once again thank them for recommending our work for publication.

Reply to referees

Forestalled Phase Separation as the Precursor to Stripe Order

Aritra Sinha, Alexander Wietek

Nature Communications submission NCOMMS-25-09251 Sinha

1. General remarks

We are grateful to the referees for their thoughtful and constructive reports. We have now made significant efforts to answer the questions raised by the third referee. Among those, the most important improvements are summarized as follows.

1. We rebut the valid concern that the clusters might only be merged stripes. This is done, among further means, by a new analysis which revealed that the holes-per-cluster can be non-integer multiples of the holes-per-stripe.
2. We give an extended size analysis on the cluster sizes, including an analysis of which fraction wraps the cylinder.
3. As our previous way of presenting cluster statistics as a plot $p_m \cdot m$ vs. m overemphasized large clusters (m being the cluster size), we now replaced all plots with p_m vs. m . As the conclusions are unchanged, we rebut the notion our analysis is an artifact of this overemphasis of larger cluster.
4. We added a large number of snapshot samples at different temperature values, which give a more accurate picture of the charge distribution.

We now start giving a more precise summary of changes, before we discuss the new results in detail. In the rest of our reply we will refer to the main text as **MT** and the Supplementary Information as **SI**.

2. Summary of major changes

- **MT Fig. 2 (METTS snapshots)**: We added legends for hole density (reference circle area) and spin correlators (reference arrow length) and the standard deviation of hole densities σ_{n_h} of the snapshot. We added a typical low temperature snapshot $T = 0.025$ at Fig. 2(d) in the main text and removed it from the Supplementary Information (SI) for better reference and clarity.
- **MT Fig. 3 (Cluster statistics)**: Main panels now show only probability p_m vs. cluster size m (stacked by *hole mass* windows). This eliminates bias toward large m . With this new hole-mass resolved histograms, we now have an answer to the appearance of commensurate peaks in the probability. The corresponding peaks occur due to **near-integer hole mass aggregation** (one-hole steps) as a contribution to growing cluster size m at intermediate T . Instead of the previous shown results shown for $(L, W) = (32, 4)$ cylinder we now choose to present data for cylinders with same size $N=96$ and different widths i.e. with configurations $(L, W) = (24, 4)$ and $(16, 6)$ and the same filling $n = 0.9375$.

- **MT Fig. 4 (Magnetism and scaling):** Now we plot the magnetic structure factor $S_m(\mathbf{k})$ at $k = (7\pi/8, \pi)$ alongside the same at $k = (\pi, \pi)$ and \bar{m} versus temperature T , showing the transfer of magnetic weight only at lower temperatures ($T \lesssim 0.06$) towards the stripe wavelength ($k = (7\pi/8, \pi)$) away from the AFM wavelength ($k = (\pi, \pi)$).
- **SI Fig. 4 (temp sweep):** Full temperature sweep of p_m vs m on 32×4 cylinder, distinguishing high- T exponential small- m , intermediate- T broad/clustering, and low- T stripe selection near $m \simeq 12$, everything with the new hole mass resolved histograms.
- **SI Fig. 10 (wrap probability):** *Wrap-around probability:* We quantify the probability of clusters which wrap around the cylinder width, P_{wrap} . We find the probability is *well below unity* in the clustering window, and it *decreases for larger width*, and approaches 1 only in the very low- T stripe regime.
- **SI Fig. 11 (length scans):** We show p_m vs m for $L=16, 24, 32, 40$ at fixed $W = 4$ and filling $n = 0.9375$. It shows additional lobes (and corresponding peaks) of hole clusters as L (and hence number of holes) grow, consistent with system-spanning fluctuations at intermediate T and they all collapse to $m \simeq 12$ at low T .
- **SI Fig. 3 (convergence and snapshots of $W=6$ cylinders):** For the wider cylinder of width $W = 6$ we now provide energy, double occupancy, and nearest neighbor spin correlator convergence vs bond dimension up to $T = 0.125$; also we have two representative snapshots with proper scales.
- Added sample script and README.md in the github METTS repository
- Added a file containing several snapshots for selected temperatures.

3. Reply to the First Referee

The authors have gone above and beyond in addressing all my concerns from the previous round of review. I am pleased to say that I have no further comments. I think this manuscript is a very valuable contribution to the literature of Hubbard model physics and I endorse its publication.

Reviewer 1 (Remarks on code availability):

It would be great if the METTS repository contained a quick guide to using the code. Currently there is no such documentation.

We thank the referee for appreciating the previous improvements suggested and for the recommendation to publish.

We have now added a sample script for METTS simulations for Fermi-Hubbard model on a cylinder and a README.md file in the METTS repository, which explains how the simulations code is used to obtain the results in the manuscript.

4. Reply to the Third Referee

Thank you for forwarding the revised paper by Sinha et al. on Forestalled Phase Separation as the Precursor to Stripe Order.

The authors included many of the suggestions in their revision which I appreciate. My main criticism, however, persists if not strengthened. Hence I am afraid, the clustering analysis needs further improvement before I can make a recommendation.

My major worries remain finite-size effects in the clustering analysis, and further questions on the data itself.

One may look at the present analysis coming from low- T : starting from stripe order, stripes start to move. As such they will get closer to each other at some places but move away from each other at other places. Based on an arbitrary but fixed algorithm to determine cluster sizes, this also will naturally lead to an increase in cluster sizes. Specifically, in the presence of finite width- W cylinders, this also would explain the pronounced commensurate peaks currently seen in the $m * p_m$ distributions, as well as the clustering wrapping around the full circumference in the snapshots such as Fig. 2. This weakens the argument on forestalled phase separation.

We thank the referee for the deep analysis and for opening up astute questions. We now tackle the challenge to resolve the three questions posed in the previous paragraph:

1. "*The commensurate peaks stem from the finite width- W cylinders.*": We thank the referee for the sharp question. The distinct peaks in the cluster-size distribution do **not** arise from finite-width commensuration. While the finite cylinder size does have a measurable effect on the cluster distribution, the origin of the "pronounced commensurate peaks" is different. Our new analysis shows that each peak is tied to an *approximately integer hole mass per cluster*. Concretely, for every cluster \mathcal{C} with size m and hole mass $\rho(\mathcal{C}) = \sum_{r \in \mathcal{C}} n_h(r)$, we coarse-grain $\rho(\mathcal{C})$ into non-overlapping windows

$$\mathcal{I} = \{[0, 0.6), [0.6, 1.6), [1.6, 2.6), [2.6, 3.6), [3.6, 4.6), [4.6, 5.6), \dots\},$$

and decompose the density-weighted distribution p_m as $p_m = \sum_{I \in \mathcal{I}} p_m^{(I)}$

$$p_m^{(I)} = \frac{\sum_{\text{snapshots } \mathcal{C}: |\mathcal{C}|=m} \sum_{\rho(\mathcal{C}) \in I} \rho(\mathcal{C})}{\sum_{\text{snapshots } \mathcal{C}} \sum \rho(\mathcal{C})}$$

Stacked histograms of p_m versus m reveal that lobe k is dominantly supplied by the window centered near k holes, $I_k \simeq [k - 1 + s, k + s)$ with $k = 1, 2, 3, \dots$. We choose $s = 0.6$ to represent the data best for our threshold at $c = 0.5$. Physically, this reflects the antiferromagnetism-assisted near-integral aggregation of holes: on cooling from the high- T dilute regime, weight shifts from single-site to larger clusters, and the emergent lobes track *integer* hole counts rather than a geometric commensuration set by W . In MT Fig. 3 and here Fig 1, we observe the same lobe-window correspondence on both 24×4 and 16×6 cylinders with the same total number of sites ($N = 96$) across $T = 2.0, 0.30, 0.25, 0.125$ (and 0.025 for $W = 4$). Thus, the phenomenon is *width-robust*: changing W shifts the mean cluster size $\bar{m} = \sum_m m p_m$ (larger for $W = 6$ at intermediate T), but does not remove or rearrange the integer-hole lobe structure.

A subtle point concerns the apparent "1-hole" coloring for stripe segments. Because our thresholding isolates stripe *cores* (roughly three sites of an ~ 8 -site wavelength), a single segment often carries

Figure 1: Stacked histograms of the density-weighted cluster-size distribution p_m vs m , colored by cluster hole mass $\rho(\mathcal{C})$, reveal width-robust integer-hole lobes.

$\rho(\mathcal{C}) \in [0.6, 1.6)$, which bins near one hole—even though the *full* stripe wavelength integrates to about two holes. This explains why segments contribute predominantly to the I_1 bin while the global picture remains consistent with near-two-hole stripes. We now write in the main text:

Finite-size effects are thus separable: increasing W at fixed area broadens p_m and raises \bar{m} ; increasing L at fixed W softens the large- m cutoff and reveals additional lobes (see SI). Importantly, in the clustering window the probability that a cluster wraps around the cylinder width remains well below unity and decreases with W (see SI). All of these are inconsistent with merely meandering stripe fluctuations at finite temperature.

In fact, we think that this extended analysis adds a very important new result to our manuscript. Thus, we thank the referee for insisting on working this out precisely. The main point of our rebuttal of the "fluctuating stripe" scenario now is the fact that cluster sizes are not only integer multiples of the number of holes in a stripe (blue tone lobes in the density distribution), but also fractional values (here half-integer, red tone lobes).

2. *"The clusters wrap around the cylinder"*: This hypothesis can be easily tested by computing the fraction of clusters wrapping around the cylinder. The new Fig. 10 in the SI and Fig. 2 here shows, that in the clustering window, P_{wrap} is well below unity; for $c=0.5$ it is typically $\lesssim 50\%$ on $W=4$ and $\lesssim 30\%$ on $W=6$, while across c values it remains $< 70\%$ ($W=4$) and $\lesssim 50\%$ ($W=6$), rising toward unity only deep in the stripe regime.
3. *"The analysis is based on an arbitrary but fixed algorithm"*: We had already investigated the dependence of our result on the threshold and doping in the now Fig. 5 of the SI. As such we kindly disagree that the previous analysis has only been performed "with an arbitrary but fixed algorithm to determine the cluster size".

Figure 2: For thresholds $c = 0.0, 0.5, 1.0$, the fraction of clusters that occupy both transverse edges P_{wrap} is plotted versus temperature on a 32×4 cylinder at $D = 2000$ (blue) and 16×6 cylinders at $D = 3000$ (orange) and $D = 4000$ (green).

The wider spread in the distributions $m * p_m$ for intermediate temperatures partly may also be seen as an artifact of the presentation of the data that is biased towards larger m by plotting $p_m - > m * p_m$.

We agree with the referee that $p_m m$ does indeed emphasize large clusters. Therefore, the main clustering plots in MT Fig. 3 now show p_m vs. m instead of $p_m m$ vs. m . Our conclusions of broad distributions at intermediate T where we see clear presence of large clusters, and a sharp low- T stripe-selection, remain unchanged.

Lastly, it is known that there is a π phase-shift across AF domains (stripes) which is only briefly mentioned in the SM, but not much addressed otherwise. Yet is it possible that due to finite- T two stripes can close the space in between them when moving closer to each other, potentially with a weak attraction due to better satisfied nearest-neighbor AF correlations?

We would like to put forward three independent observations that argue against a scenario of merely thermally wandering static stripes in the clustering window:

- **Spin structure factor at MT Fig. 4:** In the clustering regime, the dominant magnetic weight is at (π, π) ; the incommensurate $(7\pi/8, \pi)$ part (newly added) related to magnetic signature of the stripe order overtakes only at lower T (stripe regime). Thus the intermediate- T regime is not stripe-like. See MT Fig. 4 and here Fig. 3.
- **Charge structure factor at MT Fig. 5:** At intermediate T we observe an “inner” low- k peak that shifts with L and is absent in free-fermion grand-canonical and canonical baselines; the stripe peak at $(\pi/4, 0)$ appears only at low T . This is consistent with long-wavelength, interaction-driven density fluctuations (forestalled PS), not with fluctuating static stripes.
- **Integer-mass lobes at MT Fig. 3 and non-wrapping clusters at SI Fig. 10:** The cluster histograms exhibit discrete lobes corresponding to near-integer hole mass per cluster (stepwise, one-hole aggregation on an AFM background). In the same temperature window the wrap probability $P_{wrap}(T)$ remains bounded well below unity and decreases for larger width. Both features contradict a picture of stripes that simply meander and occasionally merge.

Figure 3: Density-weighted mean cluster size $\bar{m} = \sum_m p_m m$ (maroon) and magnetic structure factors $S(\pi, \pi)$ (black) and $S(7\pi/8, \pi)$ (teal) versus temperature T at $U = 10$ and $n = 0.9375$. The dashed black line is a guide to the eye where $S(\pi, \pi)$ and $S(7\pi/8, \pi)$ intersect and \bar{m} starts settling into the stripe phase to the left of that line; thus we use it as an operational definition of stripe onset. The shaded regions indicate the low-temperature stripe regime (grey) and the crossover window of forestalled phase separation (lavender), which smoothly fades into the high-temperature phase.

In more detail:

Fig. 3: the data in the newly added insets to panels (c-d) bends downwards on a log-log scale. Hence I see no reason why to fit a power law there, instead of the exponential same as in panel (a). Then with $p_m \sim \exp(-m/(W * \xi))$ with correlation length ξ along the cylinder of width $W = 4$, the data $m * p_m$ in the main panels of Fig. 3 is peaked at $m_{pk} = W * \xi$, suggesting $\xi \sim m_{pk}/W < 5 = O(W)$. Then one may phrase the data in Fig. 3 as follows: the correlation length increases up to order of the width of the cylinder before it decreases again.

We agree that our earlier use of a power-law description for the insets was not entirely appropriate; those tails are better captured by an exponential envelope. In the revised MT Fig. 3 and Fig. 1 here, we therefore only show p_m vs. m . At $T = 0.125$, the peak positions follow $m_{pk} \approx 3W$ for both geometries — $m_{pk} \approx 12$ on 24×4 and $m_{pk} \approx 18$ on 16×6 — consistent with an exponential form $p_m \sim \exp(-m/(W\xi))$ and an effective longitudinal scale $\xi \approx 3$.

We also see in SI Fig. 11 (length scans at fixed $W = 4$) that at intermediate temperatures p_m grows up to $m \approx 12$ before decaying with an exponential envelope; this behavior is robust across L . That said, a single-parameter ξ does not capture the interaction-driven structure that is central to our findings: at intermediate T the size histograms resolve into distinct lobes tied to near-integer hole mass per cluster (AFM-assisted, one-by-one aggregation), whereas at low T the distribution collapses to $m \approx 3W$ reflecting the selected stripe-core size rather than a correlation length. In short, while an exponential envelope is a good descriptor of the large- m decay on finite cylinders, the integer-lobe pattern at intermediate T is the salient physics and cannot be explained by a single ξ or by meandering of stripes.

Hence there are strong finite-size effects related to the chosen cylinder of width W on top of the period $m(8,12)(2,3)W$ that appears as a single peak in the $T > 0$ data at $m=3W$. This also appears to hold for the $W=6$ cylinder in the SM where breaking of the stripes across the circumference may be due to a difficulty converging the thermal state.

We agree that $W = 6$ cylinders are more demanding than $W = 4$. For all $W = 6$ data shown, we provide explicit convergence checks (energy, double occupancy, and NN spin correlators) versus MPS bond dimension; see the modified Fig. 3 of SI where we added an additional subplot for convergence of NN spin correlators in subplot (c). Thus, we can clearly reject the assertion that non-converged results are presented. We also note that METTS at higher temperatures requires smaller bond dimensions, which further supports the robustness of the intermediate-T clustering results at $W = 6$. This is also the reason that lowest-temperature data is only shown for the $W = 4$ cylinder.

Fig. 2: I miss absolute values in this plot: there should be a legend in terms of a particular circle that indicates a particular hole concentration $p_i = 1 - \langle n_i \rangle$. Currently, the variations seen in the hole concentration in Figs. 2 (MP) and 3, 4, 9 are only on a relative basis. I may have missed it, but I also did not see any values specified for the standard deviation σ [Eq. 4] (besides that σ is also used as spin index).

Ditto for the spin expectation values: the value corresponding to a particular length of the arrow should be given as a quantitative reference.

We thank the referee for pointing this out. We now use σ_{n_h} for the standard deviation of the hole-density distribution. We have now added a legend to each plot with reference values. Please note that the arrow length for spin correlations is often larger at higher temperatures. This is because at higher temperatures holes are localized and the origin is more likely to sit **outside** a region where holes are present and hence local correlations appear stronger.

Why are the hole distribution vs spin distribution shown on an inequivalent footing? The hole concentrations are specified as local expectation values, whereas spin expectation values which are specified as (long-range) static correlations [black cross in Figs. 2 (MP), 3, 4, and 9 (SM) at rather arbitrary locations picked inside a spin cluster]. Is it because the plain local expectation values for the spin are very small because of the thermal mixed state? If so, this deserves a brief discussion / a few comments on its own.

The open boundary conditions pin the order by prohibiting translational symmetry. Due to the intact spin rotation symmetry of the SU(2), the spins can technically point in all directions, making the projection on an S^z basis less meaningful. Hence, spin correlations are presented here. We made a comment in MT to that end where we describe Fig. 2:

Because SU(2) symmetry is intact, local moments in a thermal mixed state are small and basis dependent; we therefore visualize correlations rather than local $\langle S^z \rangle$.

There are surprisingly many sizable parallel spins in Fig. 2 in the hole clusters for the lowest temperature $T=0.1$ (panel c) (for a completely depleted site there would be no spin correlations).

This is correct, however the sites are rather far from being depleted, as can now be seen with new annotations we added upon the referee's comment. A typical value of the site-local hole density in a cluster varies from 0.1 to 0.15. Also, we would like to remark that already in the few snapshots presented initially in the main text and now in the additional snapshots no pronounced stripe correlations are found within the clusters. This observation should be appreciated with respect to the previous assertion of the referee that the clusters might be only (possibly merged) stripes.

Somehow the cluster sizes in the snap shot Fig. 2a does not seem to exceed $m < 12=3W$, whereas the respective distribution in Fig. 3c (also at $T=0.20$) suggests much larger clusters to be present. In this sense, is Fig. 2a representative (are the T-labels correct?), or does Fig. 3c overemphasize the tail by plotting $p_m - > m * p_m$?

To give a better impression of how cluster sizes are distributed, we show a large number of snapshots in the additional file, the statistics of which is presented in MT Fig. 3 and 4. The plot of $p_m m$ vs m indeed overemphasized larger clusters, which is why we replaced all plots with p_m vs m in the revised manuscript.

Looking at Fig. 2a assuming [$T > 0$ (!)]: there seems to be a horizontal period of $(2+6=8)$. With a doping of $p=1/16$ [$n=0.9375 = (1-1/16)$; why not write $p=1/16$ instead of $n=0.9375$?] this suggests a half-filled stripe. This actually may prevent two stripes from completely merging with each other while fully satisfying AF order?

We appreciate the observation and agree with the interpretation. The low-temperature pattern shows a period-8 modulation (a two to three site charge core plus the rest an AFM domain), consistent with a half-filled stripe at $n = 0.9375$ (doping $1 - n = 1/16$). In this regime, stripes act as antiphase domain walls, so bringing two stripes together frustrates the AF background; this yields an effective short-range repulsion and stabilizes a finite spacing rather than full merging—fully in line with the reviewer’s reasoning.

On notation, we retain n (filling) to avoid confusion with the probability p_m used in the cluster statistics and to remain consistent with our grand-canonical analysis, e.g., $n(\mu)$ and charge susceptibility $\chi_{\text{charge}} = \frac{\partial n}{\partial \mu}$. Where helpful, we now state the doping explicitly as $(1 - n)$.

This low- T half-filled stripe is also in precise agreement with prior DMRG studies on width-4 cylinders, which report half-filled stripes at similar dopings [Phys. Rev. Research **2**, 033073 (2020)], with full-filled stripes appearing only at substantially higher dopings or wider cylinders.

Very minor remarks:

I suggest to order the panel labels in the order of T.

Please add labels - a,b,c to Fig. 2 - a-d to Fig. SM-9

Fig. SM-6: what is the doping in panel (a), and the constant c in panel (b)?

We thank the referee for the suggestions we have now gladly implemented.

Reply to referees

Forestalled Phase Separation as the Precursor to Stripe Order

Aritra Sinha, Alexander Wietek

Nature Communications submission NCOMMS-25-09251B Sinha

We thank the referee for the careful reading and for recommending publication. The following additions in the Supplementary Information (SI) do not change our conclusions.

Summary of changes.

- Added **SI Fig. 5** (Robustness of cluster statistics at fixed $T = 0.10$): Added density weighted cluster size histograms scanning the threshold parameter c at filling $n = 0.9375$, and also probed the dependence on filling (including the requested $1/8$ doping) at fixed $c = 0.5$
- Added **SI Fig. 12** (Wrap-around probability $P_{\text{wrap}}(T)$ vs temperature) for $n \in \{0.875, 0.90625, 0.93750\}$.

1. Reply to the Third Referee

Thank you for forwarding the second revision of the paper by Sinha et al. on Forestalled Phase Separation as the Precursor to Stripe Order.

The authors put significantly more effort still into their analysis. The revised figures work well now. In particular I like the newly added color coding w.r.t. the average integer hole mass. A central concern of mine was that stripes wrap around the cylinder and thus link to finite size effects. This is now also addressed in detail in the paper and much emphasizes the original claim of the paper. With these updates in place I recommend publication in *Nature Communications*.

We thank the referee for the positive assessment and for recommending publication. Below we address the final points.

Final recommendations:

The doping of $1/16$ suggests a minimal cluster size of 16 sites to localize one dopant hole. Assuming directional invariance, such a cluster may thus be concentrated onto a circle of diameter $d \sim 4.5$. Now if the width of a cylinder is smaller than this, most clusters will actually wrap the cylinder which will manifest as ‘stripes’. E.g. this holds for the $W = 4$ cylinder in Figs. 2 or S8. On the other hand, when $W > d$ as for $W=6$ in Fig. 8 or the lower panels in Fig. 3, the wrapping of the ‘stripes’ around the cylinder will be considerably broken up into puddles (‘clusters’) as seen in Fig. S3. This suggests that for a larger doping of $1/8$ with $d \sim 3.19$, the breaking up of the stripes around the cylinder already should also be considerable for $W=4$. It would be very instructive to see this picture supported, specifically so given

Figure 1: Robustness of Cluster statistics at fixed temperature $T = 0.10$ on cylinder of size $L \times W = 32 \times 4$. In the top row we probe the sensitivity to the threshold parameter c at $n = 0.9375$ of the stacked p_m vs. m . Larger c leads to formation of clusters of lower size, however the large distribution of m at finite temperature persists and the near-integer hole mass character of the lobes survives as well. In the bottom row, we study the doping dependence at fixed $c = 0.5$. We find persistent large distribution of cluster sizes; however as filling decreases it damps the lobe structure.

that there is much literature on the Hubbard model for doping $1/8$. Based on Fig. S5, apparently such data already exists.

We agree that both geometry and doping control cluster formation and wrapping. As temperature is lowered, clusters grow and wrapping increases. However, our finite- T Markov-chain-based diagnostic identifies fluctuating, anisotropic clusters aided by antiferromagnetic (AFM) correlations, not rigid circular droplets. The transverse scale relevant for wrapping at low T is set by charge stripe correlations (stripe wavelength ~ 8 and core width ≈ 3 sites on $W = 4$), rather than by a circular diameter deduced from '16 sites per hole.' Consistent with the referee's intuition, increased hole doping reduces the likelihood of wrap-around at intermediate T . Concretely, the SI now includes:

(1) **Cluster statistics at temperature $T = 0.1$ across thresholds and fillings (SI Fig. 5 and here Fig. 1).** If we look at the bottom row, we find that as the filling is reduced from $n = 0.9375$ (doping $1/16$) to 0.875 (doping $1/8$), the antiferromagnetic background that stabilizes sharp domain walls gets weakened and consequently the oscillatory, lobe-like structure of p_m is suppressed. However, the theme of large distribution in cluster sizes still persists across a wide range of dopings thereby strengthening our primary claim.

(2) **Wrap probability vs temperature across fillings (SI Fig. 12 and here Fig. 2).** We show that, for several fillings, inside the clustering window $P_{\text{wrap}} < 1$ and is systematically lower at larger hole doping $1 - n$ for cylinder of width $W = 4$; upon further cooling $P_{\text{wrap}} \rightarrow 1$ as static stripes form. This is for $c = 0.5$. It agrees with the referee's comment that at larger doping, clusters will wrap around the cylinder less.

Fig. 2d shows 4 stripes which according to color coding in Fig. 3e should carry a hole mass of about 1 (1h). However, the system in Fig. 2 has a doping of $1/16$ and hence carries a total of $8 = 32 \cdot 4/16$ holes.

Figure 2: Probability that a cluster wraps the transverse direction, $P_{\text{wrap}}(T)$, on a 32×4 cylinder at $U = 10$ for fillings $n \in \{0.875, 0.90625, 0.93750\}$ and $c = 0.5$. In the intermediate- T clustering window $P_{\text{wrap}} < 1$ and is systematically lower at larger hole doping $1 - n$; upon further cooling, P_{wrap} increases toward unity as static stripes form.

Where does the difference of 4 holes go? Being close to the ground state, Fig. 2d rather suggests 2h per stripe. Is this discrepancy an artifact of the choice of $c=0.5$? If yes, then isn't the choice of the intervals I_n , and hence the color coding in Fig. 3 strongly sensitive on the particular choice of c ? Did the choice of $c=0.5$ happen to be an accidentally good choice in this regard? Please clarify.

The color coding denotes the integrated hole mass inside each connected component (“cluster core”) selected by the hole density mask $n_h(i) > \bar{n}_h + c\sigma_{n_h}$. Only sites inside the mask contribute to a core’s mass, $m_{\text{core}} = \sum_{i \in C} n_h(i)$; sites outside the mask are excluded. Consequently, in Fig. 2d the four wrapping cores carry $\approx 1h$ each ($\sum m_{\text{core}} \approx 4h$), while the remaining $\approx 4h$ of the total $N_h = 8$ resides outside the cores. Lowering c thickens the mask, moving weight from the background into the cores without changing the total hole count.

Sensitivity to c : In the SI we added a fixed- T scan at $T = 0.1$ (inside the forestalled phase separation regime) over $c \in \{0, 0.35, 0.70, 1.00\}$ at $n = 0.9375$ (here Fig. 1, top row). As expected, peak locations in p_m of the cluster lobes shift with c because the threshold changes how tightly we crop cores. However, the qualitative features are robust: the broad distribution of cluster sizes at intermediate T persists, and the dominant mass windows I_k continue to progress in near-integer steps as m increases. Thus, $c = 0.5$ is **not** a finely tuned choice; it is simply a balanced one such that we don’t overemphasize individual site variations yet meaningfully detect clusters.